# DOPL: Direct Online Preference Learning for Restless Bandits with Preference Feedback

**Guojun Xiong**[1]*, **Ujwal Dinesha**[2], **Debajoy Mukherjee**[2], **Jian Li**[3], **Srinivas Shakkottai**[2]
[1]Harvard University, [2]Texas A&M University, [3]Stony Brook University

## Abstract

Restless multi-armed bandits (RMAB) has been widely used to model constrained sequential decision making problems, where the state of each restless arm evolves according to a Markov chain and each state transition generates a scalar reward. However, the success of RMAB crucially relies on the availability and quality of reward signals. Unfortunately, specifying an exact reward function in practice can be challenging and even infeasible. In this paper, we introduce Pref-RMAB, a new RMAB model in the presence of *preference* signals, where the decision maker only observes pairwise preference feedback rather than scalar reward from the activated arms at each decision epoch. Preference feedback, however, arguably contains less information than the scalar reward, which makes Pref-RMAB seemingly more difficult. To address this challenge, we present a direct online preference learning (DOPL) algorithm for Pref-RMAB to efficiently explore the unknown environments, adaptively collect preference data in an online manner, and directly leverage the preference feedback for decision-makings. We prove that DOPL yields a sublinear regret. To our best knowledge, this is the first algorithm to ensure $\tilde{\mathcal{O}}(\sqrt{T \ln T})$ regret for RMAB with preference feedback. Experimental results further demonstrate the effectiveness of DOPL.

## 1 Introduction

The restless multi-armed bandits (RMAB) problem (Whittle, 1988) is a time slotted game between a decision maker (DM) and the environment. In the standard RMAB model, each "restless" arm is described by a Markov decision process (MDP) (Puterman, 1994), and evolves stochastically according to two different transition functions, depending on whether the arm is activated or not. Scalar rewards are generated with each transition. The goal of the DM is to maximize the total expected reward under an instantaneous constraint that at most $B$ out of $N$ arms can be activated at any decision epoch. Although RMAB has been widely used to study constrained sequential decision making problems (Meshram et al., 2016; Bertsimas & Niño-Mora, 2000; Yu et al., 2018; Borkar et al., 2017; Killian et al., 2021; Mate et al., 2021; 2020; Xiong et al., 2023a), it is notoriously intractable due to the explosion of state space (Papadimitriou & Tsitsiklis, 1994). To address this issue, there are extensive studies on developing low-complexity index policies (Whittle, 1988; Larrañaga et al., 2014; Bagheri & Scaglione, 2015; Verloop, 2016; Zhang & Frazier, 2021) for offline RMAB and reinforcement learning (RL) algorithms (Tekin & Liu, 2011; Cohen et al., 2014; Fu et al., 2019; Jung & Tewari, 2019; Wang et al., 2020; Nakhleh et al., 2021; Xiong et al., 2022b; Avrachenkov & Borkar, 2022; Xiong & Li, 2023; Wang et al., 2024a; Chen et al., 2024) for online RMAB.

However, the success of these index polices and RL algorithms for the standard RMAB crucially relies on the availability and quality of reward signals or reward feedback. Unfortunately, specifying an exact reward function in practice can be challenging, and obtaining the precise reward feedback may be even infeasible (Wirth et al., 2017; Casper et al., 2023). Instead, it is often much easier, faster and less expensive to collect preference feedback (e.g., provided by humans). This is especially pronounced for existing RMAB applications in online advertisement (Meshram et al., 2016) and healthcare (Killian et al., 2021; Mate et al., 2021; 2020). For example, to understand the liking for a given pair of products $(A, B)$ in online ads, it is much easier for users to answer preference-based

---

*This work was done when G. Xiong was a PhD student at Stony Brook University.

queries like: *"Do you prefer product A over B?"*, rather than the absolute counterpart, *"How much do you score products A and B in a scale of [0-10]?"*

Learning from preference feedback has gained much popularity in the machine learning community, from the famous dueling bandit problem (Yue et al., 2012) to the more recent reinforcement learning from human feedback (RLHF) (Christiano et al., 2017; Ziegler et al., 2019), e.g., for the large language model (LLM) alignment (Ouyang et al., 2022). Motivated by this and to address the limitation of the standard RMAB model, we propose a new *RMAB model in the presence of preference feedback*, dubbed as PREF-RMAB. Real-world applications that can be modeled as PREF-RMAB problems are presented in Appendix B. Like the standard RMAB, the DM in PREF-RMAB still activates a subset of $B$ arms at each decision epoch. Unlike the standard RMAB, the DM in PREF-RMAB cannot directly observe a scalar reward from each activated arm, and instead can only observe a pairwise comparison feedback from the activated subset of arms. Preference feedback, however, arguably contains less information than the scalar reward, which makes PREF-RMAB seemingly more difficult. A thorough understanding of how the preference feedback influences the DM in PREF-RMAB remains elusive. Consequently, this prompts us to pose the following question:

*Can we design efficient RL algorithms for* PREF-RMAB *with sublinear regret guarantees?*

In response to the this question, a straightforward method as inspired by RLHF is to learn a scalar reward function to represent the preference feedback of humans, and then apply existing RL algorithms for RMAB with this estimated reward function to the PREF-RMAB problem. The downside of directly applying this RLHF method to PREF-RMAB is its complexity and insufficiency. The limited theoretical analysis of RLHF (Chen et al., 2022; Zhu et al., 2023; Saha et al., 2023; Du et al., 2024) typically assumes a linear reward function, i.e., there exists a known feature mapping to specify the feature vectors of state-action pairs. However, the feature mapping in practice is often unknown and fine-tuning its dimension can be resource-intensive and prone to error. In view of these issues, there is an innovative line of work that directly learns from preferences without explicit reward modeling, such as the popular direct preference optimization (DPO) (Rafailov et al., 2024) and among others (Zhao et al., 2023; Azar et al., 2024; Ethayarajh et al., 2024; Tang et al., 2024). However, most of the latest RLHF and DPO based methods are considered offline and provided with a given and pre-collected human preference (and transition) dataset, leading to the problem of overoptimization (Xiong et al., 2023b), while the DM in PREF-RMAB often interacts with unknown environments in an online manner to model constrained sequential decision making problems in aforementioned RMAB applications.

In this paper, we consider *the online* PREF-RMAB, and show how to *directly* leverage the preference feedback to design a provably efficient RL algorithm for PREF-RMAB without explicit reward modeling. Specifically, we develop *Direct Online Preference Learning* (DOPL), an episodic RL algorithm that optimizes the same objective as the standard RMAB problem but can efficiently explore the unknown environment, adaptively collect preference data in an online manner, and directly leverage the preference feedback to make decisions for PREF-RMAB.

*First*, to handle unknown transitions of each arm, we construct confidence sets to guarantee the true ones lie in these sets with high probability (Section 3.1). *Second*, by assuming a Bradley-Terry (BT) model (Bradley & Terry, 1952), DOPL may use the results of comparison between any pair of activated arms to update the estimate of the underlying unknown preference model. However, the success of this empirical preference estimation requires DOPL to activate any arm in any state frequently, which often, in practice, are hardly feasible. To handle the unknown preference model, we design a novel *preference inference*. Our key insight here is that *although some arms in some states may not be visited frequently, we can still infer the empirical average of its preference via the other arms' empirical preference estimations*. Importantly, our preference inference not only significantly reduces the number of comparison feedback required by DOPL when directly leveraging the BT model, but also is guaranteed with a bounded error (Section 3.2). Inspired by implicit exploration (Jin et al., 2020), we further construct a biased overestimated preference estimator based on the inferred preference to further encourage exploration, which also benefits the regret characterization.

*Third*, to handle the instantaneous constraint in PREF-RMAB, we develop a low-complexity index policy (Section 3.3) for the DM to make decisions. Specifically, we first solve a relaxed problem, which turns out to be a linear programming (LP) in terms of occupancy measures, rather than PREF-RMAB itself. Our key insight here is that *we can define the objective of the LP for* PREF-RMAB *in terms of preference feedback directly*. DOPL can therefore construct an index policy on top of

the solutions to the LP. This is another key difference compared to the solutions (Whittle, 1988; Larrañaga et al., 2014; Bagheri & Scaglione, 2015; Verloop, 2016; Zhang & Frazier, 2021) to the standard RMAB, which heavily rely on the scalar reward.

We prove that DOPL achieves an $\tilde{\mathcal{O}}((2c_1 + \frac{4B}{1-D})\sqrt{2NT|\mathcal{S}|\ln 4|\mathcal{S}||\mathcal{A}|NT/\epsilon})$ regret, where $T$ is the time horizon, $\mathcal{S}$ is the state space, $D$ is the ergodicity coefficient and $c_1, \epsilon$ are some constants (see details in Section 4). We believe that ours is the first work to formally characterize the regret for PREF-RMAB. Importantly, the two terms in this regret clearly maps to the aforementioned design of DOPL, i.e., the first term with coefficient $c_1$ is the regret due to the online preference learning and the second term with coefficient $4B/(1-D)$ is caused by executing the direct index policy for PREF-RMAB. Furthermore, despite the fact that the DM in PREF-RMAB can only observe preference feedback, which contains arguably less information than scalar rewards, PREF-RMAB still achieves a sublinear regret as the standard RMAB, thanks to the novel design of online preference learning and direct index policy in DOPL.

## 2   PRELIMINARIES AND PROBLEM FORMULATION

We first provide a brief overview of the standard RMAB with scalar rewards, and then formally define the problem of RMAB with preference feedback and the online settings considered in this paper.

**Notation.** We use the calligraphic letter $\mathcal{A}$ to denote a finite set with cardinality $|\mathcal{A}|$. Let $\sigma$ be a permutation on $\mathcal{A}$, and $\sigma_s$ be the position of element $s$ in $\mathcal{A}$. We use bold letter $\mathbf{F}$ to denote a matrix with $\mathbf{F}(i, j)$ being the element of $\mathbf{F}$ in the $i$-th row and $j$-th column.

### 2.1   STANDARD RESTLESS MULTI-ARMED BANDITS WITH SCALAR REWARDS

For a standard infinite-horizon average-reward RMAB (Whittle, 1988), each "restless" arm $n \in \mathcal{N}$ is described by a unichain MDP (Puterman, 1994) $\mathcal{M}_n := (\mathcal{S}, \mathcal{A}, P_n, r_n)$, where $\mathcal{S}$ is the state space, $\mathcal{A} = \{0, 1\}$ is the binary action space, $P_n(s'|s, a) : \mathcal{S} \times \mathcal{A} \times \mathcal{S} \mapsto [0, 1]$ is the transition probability of reaching state $s'$ by taking action $a$ in state $s$, and $r_n(s, a) \in [0, 1] : \mathcal{S} \times \mathcal{A} \mapsto \mathbb{R}^+$ is *the scalar reward* associated with arm $n$ of each state-action pair $(s, a)$. At each time slot $t \in \mathcal{T} = \{1, \cdots, T\}$, the DM activates $B$ out of $N$ arms. Arm $n$ is "active" at time $t$ when it is activated, i.e., $A_n^t = 1$; otherwise, arm $n$ is "passive", i.e., $A_n^t = 0$. Let $\Pi$ be the set of all possible policies for RMAB, and $\pi \in \Pi$ is a feasible policy, satisfying $\pi : \mathcal{F}_t \mapsto \mathcal{A}^N$, where $\mathcal{F}_t$ is the sigma-algebra generated by random variables $\{S_n^h, A_n^h : \forall n \in \mathcal{N}, h \leq t\}$ (Avrachenkov & Borkar, 2022). Without loss of generality, we assume that only active arms yield rewards, i.e., $r_n(s, 0) = 0$ (Akbarzadeh & Mahajan, 2022; Xiong et al., 2022b), and hence for notational simplicity, we denote $r_n^t := r_n(S_n^t)$ in the rest of the paper. The objective of the DM is to maximize the long-term average reward subject to an "instantaneous constraint" that only $B$ arms can be activated at each time slot, i.e.,

$$\text{RMAB}: \quad \max_{\pi \in \Pi} \ J(\pi) := \liminf_{T \to \infty} \frac{1}{T} \mathbb{E}_\pi \sum_{t \in \mathcal{T}} \sum_{n \in \mathcal{N}} r_n^t, \quad \text{s.t.} \sum_{n \in \mathcal{N}} A_n^t = B, \quad \forall t \in \mathcal{T}. \quad (1)$$

### 2.2   RESTLESS MULTI-ARMED BANDITS WITH PREFERENCE FEEDBACK

In this paper, we consider the setting where the scalar reward $r_n^t$ in the standard RMAB is unobservable. Instead, the DM can only observe comparison feedback between any pair of the activated arms. We call this new *RMAB model in the presence of preference feedback* as PREF-RMAB.

Similar to the standard RMAB, the goal of the DM in PREF-RMAB is still to maximize the long-term average reward as in (1) by activating a subset of $\mathcal{N}^t \subset \mathcal{N}$ arms at each time slot $t$ with $B = |\mathcal{N}^t|$. For ease of presentation, we refer to (1) as the optimization problem that the DM needs to solve in PREF-RMAB in the rest of this paper. Unlike the standard RMAB, the DM only observes a preference feedback in the form of the Bernoulli variable $\alpha(s_m^t, s_n^t) \sim \text{Ber}(\mathbf{F}_m^n(\sigma_{s_m^t}, \sigma_{s_n^t}))$ between arms $(m, n) \in \mathcal{N}^t \times \mathcal{N}^t$ according to the underlying preference matrix $\mathbf{F} \in (0, 1)^{N|\mathcal{S}| \times N|\mathcal{S}|}$. The probability[1] that arm $m$ in state $s_m^t$ is preferred over arm $n$ in state $s_n^t$ is $\text{Pr}(\alpha(s_m^t, s_n^t) = 1) =$

---

[1]This probability measure can be extended to cases where $r_n(s, 0) \neq 0$ by incorporating the action-dependence of operator $\mathbf{F}$.

---

**Algorithm 1** Online Interactions between the DM and the PREF-RMAB Environment

---

**Require:** State space $\mathcal{S}$, action space $\mathcal{A}$, unknown transition functions $\{P_n, \forall n \in \mathcal{N}\}$, unknown preference matrix $\mathbf{F}$, and an initialized policy $\pi^1$;

1: **for** $k = 1, 2, \cdots, K$ **do**
2:     The DM determines a policy $\pi^k$ based on the collected preference feedback up to episode $k$;
3:     **for** $h = 1, 2, \cdots, H$ **do**
4:         DM activates a subset of $\mathcal{N}^h := \{A_n^{k,h} = 1\}$ arms under the instantaneous constraint;
5:         DM observes preference feedback by dueling any pair of arms in $\mathcal{N}^h$ through an oracle;
6:         Arm $n, \forall n$ moves to the next state $S_n^{k,h+1} \sim P_n(\cdot|S_n^{k,h}, A_n^{k,h})$;
7:         DM observes states $\{S_n^{k,h+1}, \forall n\}$;
8:     **end for**
9: **end for**

---

$\mathbf{F}((m-1)|\mathcal{S}| + \sigma_{s_m^t}, (n-1)|\mathcal{S}| + \sigma_{s_n^t})$. We assume that the preference feedback $\alpha(s_m^t, s_n^t)$ is drawn according to the widely-used Bradley-Terry (BT) model (Bradley & Terry, 1952), satisfying

$$\mathbf{F}_m^n(\sigma_{s_m^t}, \sigma_{s_n^t}) := \mathbf{F}((m-1)|\mathcal{S}| + \sigma_{s_m^t}, (n-1)|\mathcal{S}| + \sigma_{s_n^t}) = \frac{\exp(r_m(s_m^t))}{\exp(r_m(s_m^t)) + \exp(r_n(s_n^t))}. \quad (2)$$

Hence, the block matrix $\mathbf{F}_m^n \in \mathbb{R}^{|\mathcal{S}| \times |\mathcal{S}|}$ represents the preference between a pair of arms $(m, n)$ over all $|\mathcal{S}|$ states. The preference matrix $\mathbf{F}$ of all arms $\mathcal{N}$ over all states $\mathcal{S}$ can then be expressed compactly in terms of $\mathbf{F}_m^n$. For ease of presentation, we relegate the full expression of $\mathbf{F}$ to Appendix C.2 along with a toy example for illustration.

### 2.3 ONLINE SETTING AND LEARNING REGRET

We focus on the online PREF-RMAB setting, where the underlying dynamics (i.e., $P_n$) of each arm $n$ and the preference matrix $\mathbf{F}$ are unknown to the DM. The interaction between the DM and the PREF-RMAB environment is presented in Algorithm 1. The DM repeatedly interacts with $N$ arms in PREF-RMAB in an episodic manner. The time horizon $T$ is divided into $K$ episodes and each episode consists of $H$ decision epochs, i.e., $T = KH$. Let $\tau_k := H(k-1) + 1$ be the starting time of episode $k$, and hence $S_n^{k,H} = S_n^{\tau_{k+1}}, \forall n$. At $\tau_k$, the DM determines a policy $\pi^k$ and executes this policy for each decision epoch $h$ in this episode. Specifically, at decision epoch $h$, the DM activates a subset of $\mathcal{N}^h$ arms according to $\pi^k(\cdot|\{S_n^{k,h}, \forall n\})$ under the instantaneous constraint, i.e., $\mathcal{N}^h := \{n : A_n^{k,h} = 1, \sum_{n=1}^N A_n^{k,h} = B, \forall n \in \mathcal{N}\}$. The DM then observes a preference feedback between different pairs of arms in $\mathcal{N}^h$ through a comparison oracle. Each arm moves to the next state $S_n^{k,h+1}$ sampled from $P_n(\cdot|S_n^{k,h}, A_n^{k,h})$.

**Remark 1.** *A direct application of BT model to* PREF-RMAB *requires the DM to duel any pair of arms in $\mathcal{N}^h$, leading to a complexity of $\mathcal{O}(B^2)$. Although $B \ll N$ holds for many aforementioned RMAB applications, this amount of feedback may be impractical. A key contribution in this paper is that we develop a novel "preference inference" (Step 3 in Section 3.2), with which our* DOPL *only requires $B-1$ comparisons at each decision epoch. More details are discussed in Section 3.*

The goal of the DM is to minimize the learning regret, as defined by

$$Reg(\{\pi^k\}_{k=1}^K, T) := J(\pi^{opt}, T) - J(\{\pi^k\}_{k=1}^K, T), \quad (3)$$

where $J(\pi^{opt}, T)$ is the expected total rewards achieved by the offline optimal policy[2] $\pi^{opt}$ via solving (1) when the scalar rewards are observed by the DM, and $J(\{\pi^k\}_{k=1}^K, T)$ is the expected total rewards achieved by the online policies $\{\pi^k\}_{k=1}^K$ when the DM in PREF-RMAB can only observe preference feedback as defined in (2).

### 3 DIRECT ONLINE PREFERENCE LEARNING FOR PREF-RMAB

In this section, we show that it is possible to design an efficient RL algorithm that can efficiently explore the unknown environment, adaptively collect preference data in an online manner, and directly

---

[2]Since finding the offline optimal policy for RMAB or PREF-RMAB is intractable and inspired by existing RMAB literature (Wang et al., 2020; Xiong et al., 2022b; Wang et al., 2023), we characterize the regret with respect to the offline index policy to (1) in the presence of preference feedback.

leverage the preference feedback for the computationally intractable PREF-RMAB. Specifically, we leverage the popular UCRL-based algorithm (Jaksch et al., 2010) for the online PREF-RMAB setting, and develop an episodic RL algorithm named DOPL as summarized in Algorithm 2. There are three key components in DOPL: (1) maintaining a confidence set of the transition functions to deal with the unknown true transition functions; (2) online preference learning with comparison feedback by dueling arms to deal with unknown preference model; and (3) designing a low-complexity index policy that directly leverages the preference feedback for decision-making to ensure that the instantaneous constraint of PREF-RMAB is satisfied in each decision epoch.

---

**Algorithm 2** DOPL: Direct Online Preference Learning for PREF-RMAB

---

**Require:** Initialize $C_n^1(s,a) = 0$, and $\hat{P}_n^1(s'|s,a) = 1/|\mathcal{S}|$, $\forall n \in \mathcal{N}, s, s' \in \mathcal{S}, a \in \mathcal{A}$; the preference matrix $\tilde{\mathbf{F}}^1$ with all elements initialized to be 0.5; an initialized policy $\pi^1$;
1: **for** $k = 1, \cdots, K$ **do**
2:     Execute $\pi^k$ and construct the set of plausible transition kernels $\mathcal{P}_n^{k+1}(s,a)$ according to (4);
3:     Update the preferences matrix $\tilde{\mathbf{F}}^{k+1}$ according to Algorithm 3;
4:     Construct a direct index policy $\pi^{k+1}$ from preference feedback by solving (12);
5: **end for**

---

**Representing Scalar Rewards in terms of Preference Feedback.** Before presenting the three key components of DOPL, one key observation in this paper is that we can represent scalar rewards in terms of preference feedback for PREF-RMAB.

**Proposition 1.** *The underlying scalar reward $r_n(s)$ associated with arm $n$ in state $s$ is expressed as*

$$r_n(s) = r_\star(s_\star) + \ln \frac{\mathbf{F}_n^\star(\sigma_s, \sigma_{s_\star})}{1 - \mathbf{F}_n^\star(\sigma_s, \sigma_{s_\star})},$$

*where $\star$ is a "reference" arm, $s_\star$ is a "reference" state, and $\mathbf{F}_n^\star$ is the block matrix of preference between the reference arm $\star$ and the arm $n$ as defined in (2).*

**Remark 2.** *Proposition 1 indicates that the scalar reward of arm $n$ in state $s$ can be expressed in terms of (i) the scalar reward of the reference arm $\star$ in a reference state $s_\star$, and (ii) the preference feedback between arm $n$ and the reference arm $\star$, which is drawn from the underlying BT model as in (2). For simplicity, define the "preference-reference" term as $Q_n(s) := \ln \frac{\mathbf{F}_n^\star(\sigma_s, \sigma_{s_\star})}{1 - \mathbf{F}_n^\star(\sigma_s, \sigma_{s_\star})}$. Note that the reference arm $\star$ and reference state $s_\star$ can be any arm in $\mathcal{N}$ and any state in $\mathcal{S}$, but they will remain fixed in the system once selected. We will provide insights on how to select them in practice in our experimental evaluations in Section 5. More importantly, as we will show in Lemma 3, the objective of PREF-RMAB can be directly expressed in terms of preference feedback and is regardless of the scalar reward $r_\star(s_\star)$ associated with the reference arm and the reference state.*

**Proposition 2.** *Since $r_n(s) \in [0,1], \forall s, n$, $Q_n(s)$ is monotonically increasing in $\mathbf{F}_n^\star(\sigma_s, \sigma_{s_\star})$ and bounded as $Q_n(s) \in [-1, +1]$.*

**Remark 3.** *Similar to the scalar reward $r_n(s)$ in standard RMAB, Propositions 1 and 2 indicate that the preference of arm $n$ in state $s$ in PREF-RMAB is well represented by $Q_n(s)$. Intuitively, the higher the probability (i.e., $\mathbf{F}_n^\star(\sigma_s, \sigma_{s_\star})$ as defined in (2)) arm $n$ in state $s$ is preferred over the reference arm $\star$ and reference state $s_\star$, the larger the value of $Q_n(s)$ will be.*

### 3.1 CONFIDENCE SETS FOR TRANSITION FUNCTION

Since the true transition functions $P_n, \forall n \in \mathcal{N}$ are unknown to the DM, we maintain confidence sets via past sample trajectories. Specifically, DOPL maintains two counts for each arm $n$. Let $Z_n^{k-1}(s,a)$ be the number of visits to state-action pairs $(s,a)$ until $\tau_k$, and $Z_n^{k-1}(s,a,s')$ be the number of transitions from $s$ to $s'$ under action $a$. DOPL updates these two counts as follows: $Z_n^k(s,a) = Z_n^{k-1}(s,a) + \sum_{h=1}^H \mathbb{1}(S_n^{k,h} = s, A_n^{k,h} = a)$, and $Z_n^k(s,a,s') = Z_n^{k-1}(s,a,s') + \sum_{h=1}^H \mathbb{1}(S_n^{k,h+1} = s'|S_n^{k,h} = s, A_n^{k,h} = a)$, $\forall (s,a,s') \in \mathcal{S} \times \mathcal{A} \times \mathcal{S}$. DOPL estimates the true transition function by the corresponding empirical average as: $\hat{P}_n^k(s'|s,a) = \frac{Z_n^{k-1}(s,a,s')}{\max\{Z_n^{k-1}(s,a),1\}}$, and then defines confidence sets at episode $k$ as

$$\mathcal{P}_n^k(s,a) := \{\tilde{P}_n^k(s'|s,a), \forall s' : |\tilde{P}_n^k(s'|s,a) - \hat{P}_n^k(s'|s,a)| \le \delta_n^k(s,a)\}, \tag{4}$$

where the confidence width $\delta_n^k(s, a) = \sqrt{\frac{1}{\max\{2Z_n^{k-1}(s,a),1\}} \log\left(\frac{4|\mathcal{S}||\mathcal{A}|N(k-1)H}{\epsilon}\right)}$ is built according to the Hoeffding inequality (Hoeffding, 1994) with $\epsilon \in (0, 1)$.

**Lemma 1.** *With probability at least $1 - 2\epsilon$, the true transition functions are within the confidence sets, i.e., $P_n \in \mathcal{P}_n^k, \forall n, k$.*

## 3.2 ONLINE PREFERENCE LEARNING

From Prop 2, the preference-reference term $Q_n(s) := \ln \frac{\mathbf{F}_n^\star(\sigma_s, \sigma_{s_\star})}{1 - \mathbf{F}_n^\star(\sigma_s, \sigma_{s_\star})}$ relies on the reference arm $\star$ and reference state $s_\star$. Hence, to estimate the preference of arm $n$ in state $s$ or the value of $Q_n(s)$, DOPL only needs to learn one specific column of the preference matrix $\mathbf{F}$, i.e., $\mathbf{F}(:, (\star - 1)|\mathcal{S}| + \sigma_{s_\star})$. This corresponds to the preference between the reference state $s_\star$ of reference arm $\star$ with any arm $n \in \mathcal{N}$ in any state $s \in \mathcal{S}$. Since the pairwise preference feedback is a Bernoulli random variable, and the probability of the random variable being 1 is stored in preference matrix $\mathbf{F}$, an intuitive way is to estimate $\mathbf{F}$ with empirical dueling samples. Our online preference learning consists of four steps detailed as follows and the pseudocode is summarized in Algorithm 3 in Appendix C.3.

**Step 1: Pairwise Comparison.** At each decision epoch $h$ in episode $k$, the DM observes pairwise comparison feedback from all pairs out of the $B$ activated arms. Let $C_{ij}^k(s_i, s_j)$ be the total number of comparisons between arm $i$ at state $s_i$ and arm $j$ at state $s_j$ up to episode $k$, and $W_{ij}^k(s_i, s_j)$ be the number of times when arm $i$ at state $s_i$ is preferred over arm $j$ at state $s_j$. DOPL updates these two counts incrementally whenever there is a comparison between the two arms at each decision epoch.

**Remark 4.** *Since the preference matrices $\mathbf{F}$ and $\mathbf{F}^\mathsf{T}$ are complementary, i.e., $\mathbf{F} + \mathbf{F}^\mathsf{T} = \mathbf{1}_{N|\mathcal{S}| \times N|\mathcal{S}|}$, and their diagonal elements are all 0.5, we only need to learn either the upper triangular part or the lower triangular part of $\mathbf{F}$. Therefore, we only count $C_{ij}^k(s_i, s_j)$ and $W_{ij}^k(s_i, s_j)$ for arm $i$ at state $s_i$ and arm $j$ at state $s_j$, while $C_{ji}^k(s_j, s_i)$ and $W_{ji}^k(s_j, s_i)$ are not needed.*

**Step 2: Empirical Preference Estimation.** At $\tau_{k+1}$, DOPL estimates the true preference between arm $i$ in state $s_i$ and arm $j$ in state $s_j$ by the empirical average as: $\hat{\mathbf{F}}_i^{j,k+1}(\sigma_{s_i}, \sigma_{s_j}) = \frac{W_{ij}^{k+1}(s_i, s_j)}{C_{ij}^{k+1}(s_i, s_j)}$.

Despite that this is an unbiased estimation of $\mathbf{F}_i^j(\sigma_{s_i}, \sigma_{s_j})$, a key limitation of this empirical estimate is that its success requires DOPL to activate any arm in any state "very often". Unfortunately, this may not hold true since some arms in some states may not be favored by the DM. In particular, as discussed above, DOPL only needs to estimate the preference between the reference state $s_\star$ of reference arm $\star$ with any other arms' states. This means that the success of the above empirical estimate requires DOPL to activate the reference arm $\star$ and reference state $s_\star$ frequently, which may not be possible in practice. As a result, the preference-reference term $Q_n(s)$ may be not updated for a long time due to the preference feedback model in Section 2.2, making DOPL of low efficiency.

**Step 3: Preference Inference.** To address the above limitation, one contribution in this work is to show that although some arms may not be visited frequently, we can still infer the empirical average of its preference via the other arms' empirical preference estimations.

**Lemma 2.** *Given the empirical preference estimations $\hat{\mathbf{F}}(j, j_1)$ of $\mathbf{F}(j, j_1)$ and $\hat{\mathbf{F}}(j, j_2)$ of $\mathbf{F}(j, j_2)$ in the preference matrix $\mathbf{F}$, $\forall j, j_1, j_2 \in N|\mathcal{S}|$, if their estimations satisfy $|\hat{\mathbf{F}}(j, j_1) - \mathbf{F}(j, j_1)| \leq \delta_1$ and $|\hat{\mathbf{F}}(j, j_2) - \mathbf{F}(j, j_2)| \leq \delta_2$ with probability at least $1 - 2\epsilon$, where $\delta_1, \delta_2, \epsilon$ are some constants, then we can infer the value of $\mathbf{F}(j_1, j_2)$ as*

$$\hat{\mathbf{F}}^{inf}(j_1, j_2) = \frac{(1 - \hat{\mathbf{F}}(i, j_1))\hat{\mathbf{F}}(i, j_2)}{(1 - \hat{\mathbf{F}}(i, j_1))\hat{\mathbf{F}}(i, j_2) + (1 - \hat{\mathbf{F}}(i, j_2))\hat{\mathbf{F}}(i, j_1)}. \tag{5}$$

*In addition, we have $|\hat{\mathbf{F}}^{inf}(j_1, j_2) - \mathbf{F}(j_1, j_2)| \leq L(\delta_1 + \delta_2)$, with $L = 1.3$ being a Lipschitz constant.*

**Remark 5.** *Lemma 2 indicates that for a fixed reference arm $\star$ and reference state $s_\star$, even though DOPL may not visit arm $n$ in state $\star$ frequently (i.e., the empirical estimation in Step 2 may not be accurate), we can still infer its preference $\mathbf{F}_n^\star(\sigma_{s_n}, \sigma_{s_\star})$ once we have an "accurate" empirical preference estimation of $\mathbf{F}_m^\star(\sigma_{s_m}, \sigma_{s_\star})$ and $\mathbf{F}_n^m(\sigma_{s_n}, \sigma_{s_m})$ for any arm $m$ and states $s_n, s_m$. More importantly, the inference error can be bounded by some constants $L, \delta_1, \delta_2$. Thanks to this novel preference inference design, our DOPL only needs $B - 1$ comparison feedback in each decision*

*epoch, rather than $\mathcal{O}(B^2)$ when directly applying the BT model to* PREF-RMAB *(see Remark 1). An illustrative toy example for the preference inference is presented in Appendix C.4.*

Given the empirical preference estimation $\hat{\mathbf{F}}_n^{\star,k+1}(\sigma_{s_n}, \sigma_{s_\star})$ from Step 2 and the corresponding inferred preference $\hat{\mathbf{F}}_n^{\star,k+1,\text{inf}}(\sigma_{s_n}, \sigma_{s_\star})$ from Step 3, we define the corresponding errors as $\delta_{s_n,s_\star}$ and $\delta_{s_n,s_\star}^{\text{inf}}$, respectively, via Hoeffding inequality. Specifically, $\delta_{s_n,s_\star}$ is the confidence interval for empirical preference estimation between arm $n$ in state $s_n$ and the reference arm in reference state $s_\star$, which is determined as in (4). $\delta_{s_n,s_\star}^{\text{inf}}$ is the inferred confidence interval when direct comparison data between $s_n$ and $s_\star$ is not sufficiently available, which is determined as in Lemma 2. DOPL selects the one with a smaller error as the desired estimated preference, and denote it as $\hat{\mathbf{F}}_n^{\star,k+1}(\sigma_{s_n}, \sigma_{s_\star})$.

**Step 4: Overestimation for Exploration.** We further leverage the idea of implicit exploration (Jin et al., 2020) to further encourage exploration. Specifically, we increase $\hat{\mathbf{F}}_n^{\star,k+1}(\sigma_{s_n}, \sigma_{s_\star})$ with a bonus term $\delta$ to obtain a biased estimator, denoted as $\tilde{\mathbf{F}}_n^{\star,k+1}(\sigma_{s_n}, \sigma_{s_\star}) = \hat{\mathbf{F}}_n^{\star,k+1}(\sigma_{s_n}, \sigma_{s_\star}) + \delta$, where $\delta = \min(\delta_{s_n,s_\star}^{\text{inf}}, \delta_{s_n,s_\star})$ is the confidence level over $\hat{\mathbf{F}}_n^{\star,k}(\sigma_{s_n}, \sigma_{s_\star})$ with probability at least $1 - 2\epsilon$. It is clear that $\tilde{\mathbf{F}}_n^{\star,k+1}(\sigma_{s_n}, \sigma_{s_\star})$ is overestimating $\mathbf{F}_n^\star(\sigma_{s_n}, \sigma_{s_\star})$ with high probability. Using overestimation can be viewed as an optimism principle to encourage exploration, and is beneficial for the regret characterization (Jin et al., 2020).

## 3.3 DIRECT INDEX POLICY

$$\max_{\mu^\pi} \quad J(\pi) := \sum_{n \in \mathcal{N}} \sum_{s \in \mathcal{S}} \sum_{a \in \mathcal{A}} \mu_n(s,a) r_n(s) \tag{6}$$

It is known that solving the PREF-RMAB in (1) is computationally intractable even in the offline setting with scalar reward feedback (Papadimitriou & Tsitsiklis, 1994). To tackle this challenge and inspired by Whittle (Whittle, 1988), we first relax the in-

$$\text{s.t.} \quad \sum_{n \in \mathcal{N}} \sum_{s \in \mathcal{S}} \sum_{a \in \mathcal{A}} a\mu_n(s,a) \leq B, \tag{7}$$

$$\sum_a \mu_n(s,a) = \sum_{s'} \sum_{a'} \mu_n(s',a') P_n(s'|s,a'), \forall n, \tag{8}$$

$$\sum_{s \in \mathcal{S}} \sum_{a \in \mathcal{A}} \mu_n(s,a) = 1, \ \forall n, s, a, \tag{9}$$

stantaneous constraint, i.e., the constraint is satisfied on average rather than in each decision epoch. It turns out (Verloop, 2016; Zhang & Frazier, 2021; Xiong et al., 2022b) that this relaxed problem can be equivalently transformed into a LP (6)-(9) in terms of occupancy measures (Altman, 1999). Specifically, the occupancy measure $\mu_n(s,a) \triangleq \lim_{T \to \infty} \frac{1}{T} \mathbb{E}_\pi \sum_{t=1}^T \mathbb{1}(s_n^t = s, a_n^t = a)$ is the expected number of visits to each state-action pair $(s,a)$ of arm $n$, and $\mu^\pi$ of a stationary policy $\pi$ for $N$ controlled infinite-horizon MDPs is defined as $\mu^\pi = \{\mu_n(s,a) : \forall n, s, a\}$. It is obvious that the occupancy measure satisfies $\sum_{(s,a)} \mu_n(s,a) = 1$, and hence $\mu_n, \forall n \in \mathcal{N}$ is a probability measure. Unfortunately, we cannot solve this LP since we cannot directly observe the scalar rewards $r_n(s), \forall n, s$ in PREF-RMAB. To address this challenge, we first show that we can equivalently transform the LP (6)-(9) into a new LP in terms of preference feedback for PREF-RMAB.

**Deriving the LP's Objective.** With Lemma 1 and Lemma 2, our second key observation is that the objective of the LP for PREF-RMAB can be expressed in preference feedback directly.

**Lemma 3.** *The objective $J(\pi)$ of the LP (6)-(9) can be equivalently transformed into the following objective $J_{\text{DOPL}}(\pi)$ in terms of preference feedback:*

$$J_{\text{DOPL}}(\pi) := \max_{\mu^\pi} \quad \sum_{n \in \mathcal{N}} \sum_{s \in \mathcal{S}} \sum_{a \in \mathcal{A}} \mu_n(s,a) Q_n(s). \tag{10}$$

*Proof.* According to Lemma 1, the objective $J(\pi)$ of the LP (6)-(9) can be rewritten as

$$J(\pi) := \max_{\mu^\pi} \left[ \sum_{n \in \mathcal{N}} \sum_{s \in \mathcal{S}} \sum_{a \in \mathcal{A}} \mu_n(s,a) Q_n(s) + \sum_{n \in \mathcal{N}} \sum_{s \in \mathcal{S}} \sum_{a \in \mathcal{A}} \mu_n(s,a) r_\star(s_\star) \right]. \tag{11}$$

Due to the fact that the occupancy measure is a probability measure, i.e., $\sum_{s \in \mathcal{S}} \sum_{a \in \mathcal{A}} \mu_n(s,a) = 1$, we have that $\sum_{n \in \mathcal{N}} \sum_{s \in \mathcal{S}} \sum_{a \in \mathcal{A}} \mu_n(s,a) r_\star(s_\star) = N r_\star(s_\star)$, which is independent of policy $\mu^\pi$. Therefore, the objective $\tilde{J}(\pi)$ in (11) only depends on the first term. We denote it as $J_{\text{DOPL}}(\pi)$, which purely depends on the preference feedback drawn from the underlying BT model in (2). $\square$

**Remark 6.** *Lemma 3 indicates that we can relax the* PREF-RMAB *and then equivalently transform it into a new LP in terms of occupancy measures and the preference feedback directly.*

**Designing Direct Index Policy for PREF-RMAB.** Due to the lack of knowledge of the true transition functions (i.e., $P_n(\cdot|s,a), \forall s,a,n$) associated with each arm $n$, and the true preference $Q_n(s), \forall n, s$, we further rewrite the new LP with objective (10) as an extended LP (ELP) by leveraging the state-action-state occupancy measure $\omega_n(s,a,s') := P_n(s'|s,a)\mu_n(s,a)$ to express the confidence intervals of transition probabilities (see Section 3.1). Together with the estimated preference-reference term $\tilde{Q}_n^k(s), \forall s, n$ (see Section 3.2), the DM ends up to solve an ELP at each episode $k$ as

$$\omega^{k,*} = \arg\max_{\omega^k} \textbf{ELP}(\tilde{P}_n^k(s'|s,a), \tilde{Q}_n^k(s), \omega^k), \tag{12}$$

where $\omega^{k,*} = \{\omega_n^{k,*}(s,a,s'), \forall n \in \mathcal{N}\}$. We present more details about this ELP in Appendix C.1. However, the optimal solution $\omega^{k,*}$ to (12) is not always feasible for PREF-RMAB due to the fact that the instantaneous constraint in PREF-RMAB must be satisfied in each decision epoch rather than on the average sense as in this ELP. Inspired by Verloop (2016); Zhang & Frazier (2021); Xiong et al. (2022b), we further construct an index policy on top of $\omega^{k,*}$ that is feasible for PREF-RMAB. Specifically, the index assigned to arm $n$ in state $s_n^t = s$ at time $t$ is defined as $\mathcal{I}_n^k(s) := \frac{\sum_{s'\in\mathcal{S}} \omega_n^{k,*}(s,1,s')}{\sum_{a\in\mathcal{A}, s'\in\mathcal{S}} \omega_n^{k,*}(s,a,s')}$. We call this the *direct index* since it is constructed by solving a LP in preference feedback directly. We then rank all arms according to their direct indices in a non-increasing order, and activate the set of $B$ highest indexed arms at each decision epoch. We denote the resultant direct index policy as $\pi^k$ and execute it in episode $k$. Asymptotic optimality of this index policy relies on the existence of a uniform global attractor (Gast et al., 2024), a standard assumption in the RMAB literature.

# 4 THEORETICAL GUARANTEE

Now we provide the performance guarantee for DOPL, which is formally stated in Theorem 1.

**Theorem 1.** *With probability at least $1 - 2\epsilon$, the regret of DOPL satisfies*

$$Reg(T) \leq \tilde{\mathcal{O}}\left(\left(2c_1 + \frac{4B}{1-D}\right)\sqrt{2NT|\mathcal{S}|\ln\frac{4|\mathcal{S}||\mathcal{A}|NT}{\epsilon}}\right), \tag{13}$$

*where $c_1 =: \max_{i,j} \frac{2}{\mathbf{F}(i,j)(1-\mathbf{F}(i,j))}$ is a constant, and $D$ is the ergodicity coefficient (see the detailed definition in Appendix E.3).*

**Remark 7.** *The regret contains two terms. The first term with coefficient $c_1$ is the regret due to the online preference learning (Section 3.2). The second term with coefficient $4B/(1-D)$ is caused by executing the direct index policy for PREF-RMAB (to satisfy the instantaneous constraint) (Section 3.3). Clearly, the regret of DOPL is in the order of $\tilde{\mathcal{O}}(\sqrt{T\ln T})$, which matches that of the standard RMAB with scalar rewards (Ortner et al., 2012; Wang et al., 2020; Xiong et al., 2022a). However, comparing to the standard RMAB with scalar rewards, our PREF-RMAB is a harder problem in which the DM can only observe preference feedback, which arguably contains less information. This challenging setting thus requires us to develop an effective online preference learning algorithm to handle preference feedback, as well as a new index policy that can directly leverage the preference feedback to make decisions.*

**Proof Sketch.** Compared to the standard RMAB, the key challenges of our DOPL for PREF-RMAB come from the online preference learning and the direct index policy to deal with preference feedback. To address these challenges, we first decompose the regret of DOPL correspondingly.

**Lemma 4.** *Let $\{\tilde{\pi}^k, \forall k\}$ be the direct index policy executed in each episode with transition functions drawn from the confidence sets defined in (4), and the preference overestimated by Algorithm 3. Let $J(\{\tilde{\pi}^k, \forall k\}, \{\hat{\mathbf{F}}^k, \forall k\}, T)$ and $J(\{\tilde{\pi}^k, \forall k\}, \mathbf{F}, T)$ be the total rewards achieved by policy $\{\tilde{\pi}^k, \forall k\}$ with overestimated preference $\{\hat{\mathbf{F}}^k, \forall k\}$ and the true preference $\mathbf{F}$, respectively. The regret of DOPL can be decomposed and bounded as*

$$Reg(\{\pi^k\}_{k=1}^K, T) \leq \underbrace{J(\{\tilde{\pi}^k, \forall k\}, \{\hat{\mathbf{F}}^k, \forall k\}, T) - J(\{\tilde{\pi}^k, \forall k\}, \mathbf{F}, T)}_{Term_1}$$

$$+ \underbrace{J(\{\tilde{\pi}^k, \forall k\}, \mathbf{F}, T) - J(\{\pi^k, \forall k\}, \mathbf{F}, T)}_{Term_2}. \tag{14}$$

Specifically, $Term_1$ is the regret due to the preference estimation error between the true preference $\mathbf{F}$ and estimated ones $\tilde{\mathbf{F}}^k, \forall k$ (Section 3.2). $Term_2$ is the performance gap between the policy $\{\tilde{\pi}^k, \forall k\}$ in the optimistic plausible MDP and the learned index policy $\{\pi^k, \forall k\}$ for the true MDP (Section 3.3). The inequality holds due to the optimism brought by upper confidence ball search (Section 3.1) and the preference overestimation (Section 3.2), and the fact that the LP in (6)-(9) provides an upper bound for the optimal solution of PREF-RMAB in (1) (Section 3.3).

**Bounding $Term_1$.** We first bound $Term_1$, i.e., the regret caused by online preference learning.

**Lemma 5.** *With probability $1 - 2\epsilon$, we have $Term_1 \leq 2c_1 \sqrt{2N^2 T |\mathcal{S}| \ln \frac{4|\mathcal{S}||\mathcal{A}|NT}{\epsilon}}$.*

Bounding $Term_1$ requires us first to characterize the impact on the preference-reference term $Q_n(s), \forall n, s$ made by the preference estimation error, and then to bound the inner product $\sum_{k=1}^{K} \sum_{n=1}^{N} \langle \omega_n^k, \tilde{Q}_n^k - Q_n^k \rangle$, with $\omega_n^k$ and $Q_n^k$ being the vectors containing all $\{\omega_n^k(s,a,s'), \forall s, a, s'\}$ and $\{Q_n^k(s), \forall s\}$, respectively.

**Bounding $Term_2$.** Next we bound $Term_2$, i.e., the regret due to the direct index policy.

**Lemma 6.** *With probability $1 - 2\epsilon$, we have $Term_2 \leq \frac{4B}{1-D} \sqrt{2N^2 T |\mathcal{S}| \ln \frac{4|\mathcal{S}||\mathcal{A}|NT}{\epsilon}}$.*

Bounding $Term_2$ requires us to decompose the regret into each episode. The key is to leverage the ergodicity of the underlying MDP of restless arms and bound the gap between stationary distribution and the true number of visits of each state-action pair, related to the instantaneous constraint $B$.

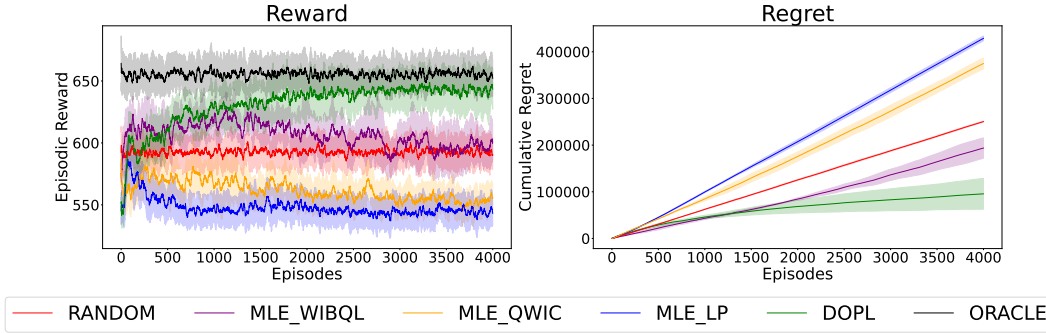

Figure 1: Comparisons of episodic reward and cumulative regret under App-Marketing environment.

## 5 EXPERIMENTS

In this section, we evaluate the efficacy of DOPL for PREF-RMAB via real-world applications. The experiment code is available at `https://github.com/flash-36/DOPL`.

**Baselines.** We consider three classes of baselines: (i) *Oracle*, which is the optimal index policy (Verloop, 2016; Zhang & Frazier, 2021), derived from solving (6)-(9) using full knowledge of the reward function and transition kernels, serving as a performance upper bound. (ii) *MLE based algorithms*, which employ maximum likelihood estimation (MLE) under the BT model to estimate rewards and then apply different learning algorithms, including *MLE_WIBQL* that utilizes a Whittle index-based Q-learning algorithm (Avrachenkov & Borkar, 2022), *MLE_LP* that estimates transition kernels from observed state transitions and solves the linear program (6)-(9) for index policy derivation, and *MLE_QWIC* that integrates MLE with the Q-learning algorithm from Fu et al. (2019). (iii) *RANDOM*, in which the DM activates each arm randomly, without preference feedback.

**Synthetic Environment.** We consider a synthetic environment named *App Marketing* that mimics user engagement and retention dynamics. A detailed description is provided in Appendix B.1 and the transition kernel and latent reward function specifications are provided in Appendix F.1.

**Real-world Environments.** We consider two real-world applications for PREF-RMAB: (i) *Continuous Positive Airway Pressure (CPAP) Therapy*. The CPAP environment (Herlihy et al., 2023; Li & Varakantham, 2022; Wang et al., 2024a) addresses patient adherence to treatment for obstructive sleep apnea. A detailed description is provided in Appendices B.2 and F.2. The objective is to maximize

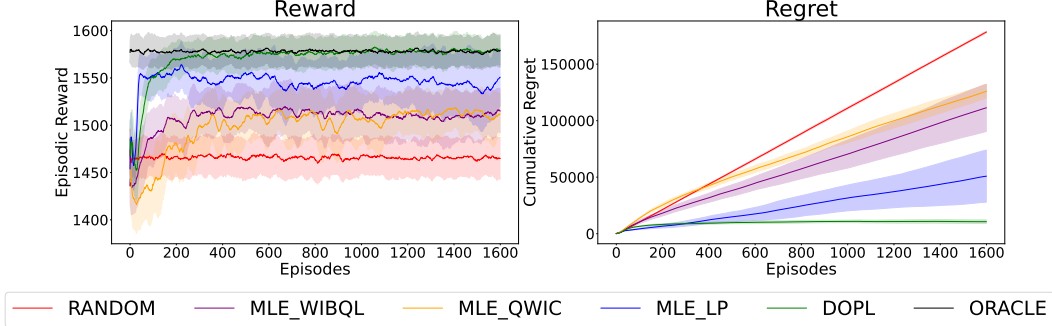

Figure 2: Comparisons of episodic reward and cumulative regret under CPAP environment.

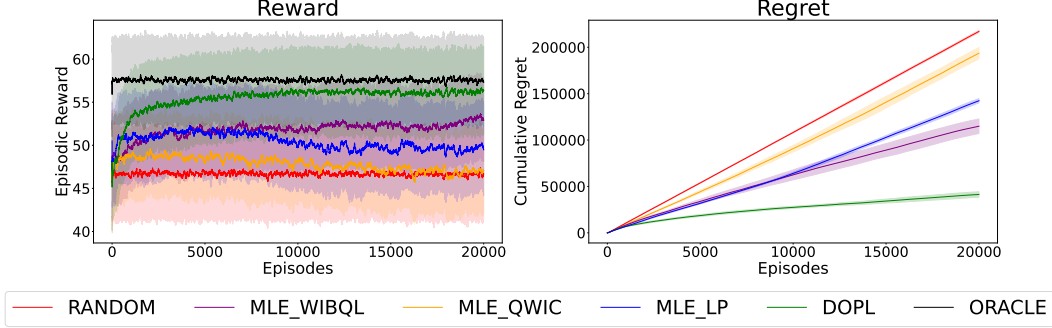

Figure 3: Comparisons of episodic reward and cumulative regret under ARMMAN environment.

cumulative patient adherence and we measure this by tracking these latent rewards. (ii) *Maternal Healthcare.* ARMMAN (Biswas et al., 2021; Killian et al., 2022) addresses a maternal healthcare intervention problem via a binary-action RMAB. For more details refer Appendices B.3 and F.3.

**Observations.** As shown in Figures 1 2 3 (left), DOPL significantly outperforms all considered baselines and reaches close to the oracle. In addition, DOPL is much more computationally efficient since it can *directly* leverage the preference feedback for decision making, while the considered baselines require a reward estimation step via MLE in the presence of preference feedback, which can be computationally expensive. Finally, our DOPL yields a sublinear regret as illustrated in Figures 1 2 3 (right), consistent with our theoretical analysis (Theorem 1), while none of these baselines yield a sublinear regret in the presence of preference feedback.

## 6 CONCLUSION

In this paper, we presented a novel restless bandits model, dubbed as PREF-RMAB, in the presence of preference feedback, and proposed DOPL, an episodic reinforcement learning algorithm that can efficiently explore the unknown environment, adaptively collect preference data in an online manner, and directly leverage the preference feedback for decision-making. We proved that DOPL achieves a sublinear regret and demonstrated its utility via experiments in real-world applications.

### LIMITATIONS

While the proposed study on the restless multi-armed bandits (RMAB) introduces a novel approach by incorporating preference feedback instead of scalar rewards, it has several limitations. The shift to preference feedback, although potentially more accessible than rewards, inherently provides less information, which implies that greater amounts of data are needed for online training. Most existing RL algorithms and preference learning methods, such as RLHF and DPO, are typically designed for offline settings with pre-collected datasets, whereas the proposed RMAB with preference feedback (PREF-RMAB) model operates in an online context. The empirical estimation of preferences, crucial for the proposed Direct Online Preference Learning (DOPL) algorithm, requires frequent activation of arms in various states for online comparisons, which would require the development of a platform for exercising options and collecting feedback from users as the algorithms learn.

ACKNOWLEDGMENTS

Research was funded in part by grants NSF CNS 2148309, 2312978 and 2337914. All all opinions expressed are those of the authors, and need not represent those of the sponsoring agencies

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

## A  RELATED WORK

**Dueling Bandit.** Dueling bandit generalizes the classic multi-armed bandit problem by pulling two arms to duel at a time and only receives the preference feedback (Yue et al., 2012). Introduced by Yue et al. (2012), the initial algorithms such as Interleaved Filter laid the groundwork for this field. Subsequent developments include the Double Thompson Sampling (DTS) (Wu & Liu, 2016), which improved empirical performance and robustness. Theoretical advancements have been made in Komiyama et al. (2015) which analyzed regret bounds. Dueling bandit has been extensively studied, under various objectives and generalizations, and applied to various applications (see Saha & Gaillard (2022) for detailed discussions). Recently, Saha & Gaillard (2022) made notable contributions by proposing a reduction from dueling bandits to multi-armed bandits, enabling optimal performance in both stochastic and adversarial environments. Their algorithm achieves an optimal regret bound against the Condorcet-winner benchmark and demonstrates robustness under adversarially corrupted

preferences. Another significant work Saha & Krishnamurthy (2022) addresses the contextual dueling bandit problem, focusing on regret minimization under realizability. Their algorithm achieves optimal regret rates for a new performance measure called best response regret, resolving an open problem regarding oracle-efficient, regret-optimal algorithms for contextual dueling bandits posed by Dudík et al. (2015). These advancements highlight the progress in dueling bandit research.

**Reinforcement Learning from Human Feedback.** Reinforcement Learning from Human Feedback (RLHF) integrates human preferences into the RL framework to guide the agent's learning process, overcoming the challenge of specifying appropriate reward functions. A pioneering work by Christiano et al. (2017) demonstrated that using human comparisons to train agents significantly improved their performance in complex tasks. Building on this, Ibarz et al. (2018) combined demonstrations with human feedback, accelerating the learning process and reducing the amount of required feedback. Most existing RLHF work (Chen et al., 2022; Zhu et al., 2023; Du et al., 2024) typically assumes a linear reward function, i.e., there exists a known feature mapping to specify the feature vectors of state-action pairs. Later on, a line of work proposed algorithms that directly learn from preferences without explicit reward modeling (Rafailov et al., 2024; Zhao et al., 2023; Azar et al., 2024; Ethayarajh et al., 2024; Tang et al., 2024). Need to mention that most of the latest RLHF and DPO based methods are considered offline and provided with a given and pre-collected human preference (and transition) dataset, leading to the problem of overoptimization (Xiong et al., 2023b). Recently, Saha et al. (2023) proposed Dueling RL, which utilizes binary preferences over trajectory pairs to guide learning, providing robust performance in environments with non-Markovian rewards and unknown transition dynamics. These advancements highlight the potential of RLHF in creating more adaptable and user-aligned reinforcement learning systems (Rafailov et al., 2024). There is also a line of research from the theoretical perspective of RLHF, showing the efficiency of RLHF comparable to conventional RL. Please see Wang et al. (2024b) for detailed discussions.

**Restless Multi-Armed Bandits.** RMAB was first introduced in Whittle (1988), and has been widely studied, see Xiong et al. (2022b) and references therein. In particular, RL algorithms have been proposed for RMAB with unknown transitions. Colored-UCRL2 is the state-of-the-art method for online RMAB with $\tilde{\mathcal{O}}(\sqrt{T \ln T})$ regret. To address the exponential computational complexity of colored-UCRL2, low-complexity RL algorithms have been developed. For example, Wang et al. (2020) proposed a generative model-based algorithm with $\tilde{\mathcal{O}}(T^{2/3})$ regret, and Xiong et al. (2022a;b) designed index-aware RL algorithms for both finite-horizon and infinite-horizon average reward settings with $\tilde{\mathcal{O}}(\sqrt{T})$ regret. However, the aforementioned existing literature on RMAB focus on the stochastic setting, where the reward functions are stochastically sampled from a fixed distribution, either known or unknown, and at each decision epoch, the reward is observed by the decision marker. To our best knowledge, this work is the first to study RMAB in the presence of preference feedback, where the decision marker only observes pairwise preference signals.

# B   MOTIVATING EXAMPLES FOR PREF-RMAB

In the following, we provide some motivating examples that can be modeled as a PREF-RMAB problem, i.e., the RMAB problem in the presence of preference feedback.

## B.1   APP MARKETING

**Scenario**: Imagine you're a marketing manager for a popular mobile app that offers various features to users. You're tasked with the job of maximizing user engagement and retention. Each day, you have a limited budget to send personalized push notifications or in-app messages to a subset of users to boost engagement and retention. Due to resource constraints, you can only target a fixed number of users daily.

**State Modelling**: Each user is modeled as a "restless" arm whose engagement state evolves over time based on whether they receive targeted marketing efforts.

$s_1$ : User at risk of churn—has not used the app in over a month.

$s_2$ : Low engagement—has not used the app in over a week.

$s_3$ : Moderately engaged—uses the app a few times a week.

$s_4$ : Highly engaged—uses the app daily, interacts with multiple features.

**Preference Feedack**: Instead of directly measuring the exact engagement metrics (which can be noisy or delayed), you can obtain preference feedback by comparing the relative engagement levels among the users you targeted. For example, after sending out notifications, you observe which users opened the app, how much time they spent, or how they interacted with the app features.

This preference feedback is modeled using the BT model:

$$\Pr(\text{User A engages with the app more than User B} \mid s_A, s_B) = \frac{\exp(r_A(s_A))}{\exp(r_A(s_A)) + \exp(r_B(s_B))}$$

where $r_A(s_A)$ and $r_B(s_B)$ represent the latent engagement and retention potentials of Users A and B based on their states.

**Objective**: The goal is to maximize overall user engagement and retention without needing to quantify the exact contribution of each user (which can be challenging due to data privacy concerns or measurement noise). By focusing on pairwise preference feedback among the targeted users, you can iteratively improve your targeting strategy to select users who are more likely to respond positively to your marketing efforts.

## B.2 CPAP TREATMENT

**Scenario**: You are a healthcare provider managing a large cohort of patients diagnosed with obstructive sleep apnea (OSA). The standard treatment for OSA is Continuous Positive Airway Pressure (CPAP) therapy, which requires patients to use a CPAP machine during sleep every night. However, patient adherence to CPAP therapy is often low due to discomfort, inconvenience, or lack of immediate perceived benefits. With limited resources (e.g., time, staff), you can provide personalized follow-up support or interventions to only a select number of patients each week to encourage adherence.

**State Modelling**: Each patient is modeled as a "restless" arm whose adherence state evolves over time based on whether they receive follow-up support. The state space for each patient considers various factors affecting adherence:

$s_1$ : Low Adherence—Uses CPAP sporadically with short usage durations.

$s_2$ : Moderate Adherence—Uses CPAP most nights but not consistently for the full duration.

$s_3$ : High Adherence—Uses CPAP every night for the recommended duration ( 4 hours per night).

**Preference Feedack**: Quantifying exact adherence levels for each patient can be challenging. Instead, you obtain preference feedback by comparing relative adherence among the patients you have intervened with.

This preference feedback is modeled using the BT model:

$$\Pr(\text{Patient A is more adherent than Patient B} \mid s_A, s_B) = \frac{\exp(r_A(s_A))}{\exp(r_A(s_A)) + \exp(r_B(s_B))}$$

where $r_A(s_A)$ and $r_B(s_B)$ represent the latent adherence potentials of Patients A and B based on their current states.

**Objective**: The goal is to maximize overall patient adherence to CPAP therapy without needing precise adherence measurements for each individual. By focusing on pairwise preference feedback among the patients you have provided interventions to, you aim to:

- Optimize Resource Allocation: Identify and support patients who are more likely to improve their adherence with additional intervention.
- Improve Health Outcomes: Enhance the effectiveness of CPAP therapy across the patient population by increasing overall adherence rates.

## B.3 MATERNAL HEALTHCARE

**Scenario**: You are collaborating with ARMMAN, a non-governmental organization focused on improving maternal and child health in underserved communities by delivering vital health information

through mobile technology. The program involves sending automated voice calls to expecting and new mothers, providing them with timely healthcare guidance. However, due to limited resources such as call center capacity and funding, you can only reach out to a subset of mothers each day. The challenge is to select which mothers to contact to maximize overall engagement with the program, thereby enhancing health outcomes.

**State Modelling**: Each mother(or benefeciary) is modeled as a "restless" arm in a Markov Decision Process (MDP) with three ordered states representing their level of engagement:

$s_1$ : Lost Cause—beneficiaries listening to less than 5% of the content of the automated calls.

$s_2$ : Persuadable—beneficiaries listening to between 5% and 50% of the content of the automated calls.

$s_3$ : Self-motivated—beneficiaries listening to more than 50% of the content of the automated calls.

**Preference Feedack**: Having to measure and keep track of the exact engagement levels of each beneficiary can be tedious or even infeasible with scale. Thus you could just rank the engagement levels among the beneficiaries you reached out to in order to get your preference feedback signal.

This preference feedback is modeled using the BT model:

$$\text{Pr(Beneficiary A is more engaged than Beneficiary B} \mid s_A, s_B) = \frac{\exp(r_A(s_A))}{\exp(r_A(s_A)) + \exp(r_B(s_B))}$$

where $r_A(s_A)$ and $r_B(s_B)$ represent the latent engagement potentials of Beneficiaries A and B based on their states.

**Objective**: The goal is to maximize overall beneficiary engagement and retention without needing to quantify the exact contribution of each user (which can be challenging due to data privacy concerns or measurement noise). By focusing on pairwise preference feedback among the targeted users, you can iteratively improve your targeting strategy to select users who are more likely to respond positively to your marketing efforts.

### B.4 SPORTS PSYCHOLOGIST

**Scenario**: Say you're a sports psychologist at a top-tier football club in the premier league and in a workday you can counsel at most 5 players. You need to choose which 5 players from the squad to counsel such that the overall team performance is maximized. Each player is modeled as a "restless" arm, whose mental and physical state evolves with time based on whether the player received counsel or not.

**State Modelling**: The following is one possible simplified state space definition that takes into account various factors that might affect player performance such as amount of sleep had last night, calorie intake, mental sharpness, sprint speed:

$s_1$ : Over 8 hrs of sleep; Met calorie intake level; Feeling mentally sharp; Sprint speed within 10% of personal best for the season.

$s_2$ : Less than 8 hrs of sleep; Met calorie intake level; Feeling mentally sharp; Sprint speed within 10% of personal best for the season.

$\vdots$

$s_{16}$ : Less than 8 hrs of sleep; Did not meet calorie intake level; Not feeling mentally sharp; Sprint speed less than 90% of personal best for the season.

**Preference Feedack**: The sports psychologist can rank the players he counselled based on their performance levels on the next game/training session in order to get the preference feedback signal needed for decision making.

This preference feedback is modeled using the BT model:

$$\text{Pr(Player A performs better than Player B} \mid s_A, s_B) = \frac{\exp(r_A(s_A))}{\exp(r_A(s_A)) + \exp(r_B(s_B))}$$

where $r_A(s_A)$ and $r_B(s_B)$ represent the latent measures of how much a player contributes to team performance.

**Objective**: Here the objective of maximizing team performance is attained without having to specifically identify how much each player contributed to the team performance (which can be a difficult thing to quantify). The sports psychologist only needs to assess which player contributed more in comparison with one another to achieve the same end goal.

# C MISSING DETAILS FOR THE MAIN PAPER

## C.1 EXTENDED LP

The detailed formulation of **ELP** is provided as follows.

$$
\mathbf{ELP}(\{\tilde{P}_n^k(s'|s,a)\}, \{\tilde{Q}_n^k(s)\}, \omega^k) : \max_{\omega^k} \; \sum_{n=1}^N \sum_{(s,a)\in\mathcal{S}\times\mathcal{A}} \sum_{s'\in\mathcal{S}} \omega_n^k(s,a,s')\tilde{Q}_n^k(s)
$$

$$
\text{s.t.} \; \sum_{n=1}^N \sum_{(s,a)\in\mathcal{S}\times\mathcal{A}} \sum_{s'\in\mathcal{S}} a\omega_n^k(s,a,s') \leq B,
$$

$$
\sum_{(s,a)\in\mathcal{S}\times\mathcal{A}} \sum_{s'\in\mathcal{S}} \omega_n^k(s,a,s') = 1, \quad \forall n \in \mathcal{N}
$$

$$
\sum_{(s',a)\in\mathcal{S}\times\mathcal{A}} \omega_n^k(s,a,s') = \sum_{(s',a')\in\mathcal{S}\times\mathcal{A}} \omega_n^k(s',a',s), \quad \forall n \in \mathcal{N},
$$

$$
\frac{\omega_n^k(s,a,s')}{\sum_y \omega_n^k(s,a,y)} - (\hat{P}_n^k(s'|s,a) + \delta_n^t(s,a)) \leq 0,
$$

$$
-\frac{\omega_n^k(s,a,s')}{\sum_y \omega_n^k(s,a,y)} + (\hat{P}_n^k(s'|s,a) - \delta_n^t(s,a)) \leq 0. \tag{15}
$$

## C.2 PREFERENCE MATRIX AND EXAMPLE

In the following, we first present the detailed compact form of the preference matrix **F** and then provide an illustrating example.

The compact form of the preference matrix $\mathbf{F} \in (0,1)^{N|\mathcal{S}|\times N|\mathcal{S}|}$ is given as:

$$
\overbrace{\qquad\qquad\qquad}^{N|\mathcal{S}|\times N|\mathcal{S}|}
$$

$$
\mathbf{F} = \begin{bmatrix} \mathbf{F}_1^1 & \mathbf{F}_1^2 & \cdots & & \cdots & & \mathbf{F}_1^N \\ \vdots & \mathbf{F}_2^2 & \mathbf{F}_2^3 & & \cdots & & \vdots \\ & & & & \cdots & & \\ \vdots & \cdots & \ddots & \mathbf{F}_i^j = \begin{bmatrix} \frac{\exp(r_i(s_1))}{\exp(r_i(s_1))+\exp(r_j(s_1))} & \cdots & \frac{\exp(r_i(s_1))}{\exp(r_i(s_1))+\exp(r_j(s_{|\mathcal{S}|}))} \\ \vdots & \ddots & \vdots \\ \frac{\exp(r_i(s_{|\mathcal{S}|}))}{\exp(r_i(s_{|\mathcal{S}|}))+\exp(r_j(s_1))} & \cdots & \frac{\exp(r_i(s_{|\mathcal{S}|}))}{\exp(r_i(s_{|\mathcal{S}|}))+\exp(r_j(s_{|\mathcal{S}|}))} \end{bmatrix} & & \vdots \\ \vdots & \cdots & \ddots & & \cdots & & \vdots \\ \mathbf{F}_N^1 & \cdots & \cdots & & \cdots & & \mathbf{F}_N^N \end{bmatrix},
$$

which contains $N \times N$ block matrix $\mathbf{F}_m^n \in \mathbb{R}^{|\mathcal{S}|\times|\mathcal{S}|}, \forall m,n \in \mathcal{N}$. In particular, the preference of arm $m$ at state $s_m$ over arm $n$ at state $s_n$ lies in the $((m-1)|\mathcal{S}|+\sigma_{s_m})$-th row and $((n-1)|\mathcal{S}|+\sigma_{s_n})$-th column of $\mathbf{F}$, i.e.,

$$
\mathbf{F}((m-1)|\mathcal{S}|+\sigma_{s_m}, (n-1)|\mathcal{S}|+\sigma_{s_n}) = \frac{\exp(r_m(s_m))}{\exp(r_m(s_m)+\exp(r_n(s_n))}. \tag{16}
$$

For ease of understanding, we present a toy example as follows.

**Example 1.** *An illustrative example with 2 arms (i.e., $N = 2$) is shown in (17), where each arm has 2 states (i.e., $|\mathcal{S}| = 2$) and rewards are denoted as $r_1(s_1), r_1(s_2)$, and $r_2(s_1), r_2(s_2)$, respectively. Given the preference matrix in (17), we have a reward order as $r_1(s_1) < r_2(s_1) < r_1(s_2) \leq r_2(s_2)$.*

*Note that the comparison between the same arm and the same state always results in* $0.50$ *according to the Bradley-Terry model in (2).*

$$
\mathbf{F} = \begin{array}{c} \\ r_1(s_1) \\ r_1(s_2) \\ r_2(s_1) \\ r_2(s_2) \end{array}
\begin{array}{cccc} r_1(s_1) & r_1(s_2) & r_2(s_1) & r_2(s_2) \\ \begin{bmatrix} 0.50 & 0.30 & 0.33 & 0.24 \\ 0.70 & 0.50 & 0.54 & 0.43 \\ 0.67 & 0.46 & 0.50 & 0.39 \\ 0.76 & 0.57 & 0.61 & 0.50 \end{bmatrix} \end{array} \tag{17}
$$

According to Example 1, the preference of arm 2 at state $s_1$ over the arm 1 at state $s_2$ can be located at the 3rd row and the 2nd column of $\mathbf{F}$, i.e., $\mathbf{F}(3,2)$, where $\sigma_{s_1} = 1$ and $\sigma_{s_2} = 2$.

### C.3 Pseudocode of Online Preference Learning

The pseudocode of online preference learning in Section 3.2 is presented as the following algorithm. In particular, we focus on the detailed procedures of the algorithm at the $k$-th episode, $\forall k$.

---

**Algorithm 3** Online Preference Learning for the $k$-th Epsiode

---

**Require:** Arms set $\mathcal{N}$, confidence parameter $\epsilon \in (0,1)$, reference arm $\star$ and reference state $s_\star$;
**Ensure:** Estimating the true $\mathbf{F}$;
1: Initialize: $C_{ij}^k(s_i, s_j) = C_{ij}^{k-1}(s_i, s_j)$ and $W_{ij}^k(s_i, s_j) = W_{ij}^{k-1}(s_i, s_j), \forall i, j \in \mathcal{N}, s_j, s_j \in \mathcal{S}$;
2: **for** $h = 1, 2, \ldots, H$ **do**
3:     At each time step, choose $B$ arms denoted as $\mathcal{A}^h$ and perform $(B-1)$ duels randomly;
4:     **for** $i, j \in \mathcal{A}^h$ do **do**
5:         $C_{ij}^k(s_i, s_j) \leftarrow C_{ij}^k(s_i, s_j) + 1$;
6:         Observe $\alpha_h(i \text{ over } j) = 1 - \alpha_h(j \text{ over } i)$;
7:         Define $\mathbb{1}_h(i,j) := \mathbb{1}(\alpha_h(i \text{ over } j))$ and $W_{ij}^k(s_i, s_j) \leftarrow W_{ij}^k(s_i, s_j) + \sum_{\tau=1}^{h} \mathbb{1}_\tau(i,j)$;
8:     **end for**
9: **end for**
10: Empirical estimation of each element in $\mathbf{F}$ is calculated according to $\hat{\mathbf{F}}_i^{j,k}(\sigma_{s_i}, \sigma_{s_j}) := \frac{W_{ij}^k(s_i, s_j)}{C_{ij}^k(s_i, s_j)}$
     with estimation error $\delta_{s_i, s_j}, \forall \sigma, j \in \mathcal{N}, s_i, s_j \in \mathcal{S}$;
11: Select one element $j$ from the $((\star - 1)|\mathcal{S}| + \sigma_{s_\star})$-th column of matrix $\hat{\mathbf{F}}^k$m i.e., $\hat{\mathbf{F}}^k(j, ((\star - 1)|\mathcal{S}| + \sigma_{s_\star}))$ with the smallest estimation error $\delta$;
12: Infer $\hat{\mathbf{F}}_n^{\star,k+1,\text{inf}}(\sigma_{s_n}, \sigma_{s_\star})$ with $\hat{\mathbf{F}}^k(j, ((\star - 1)|\mathcal{S}| + \sigma_{s_\star}))$ according to (5) and keep the corresponding inference error $\delta_{s_n, s_\star}^{\text{inf}} \; \forall n \in \mathcal{N}, s_n \in \mathcal{S}$;
13: DOPL selects the one with a smaller error as the desired estimated preference, and denote it as $\hat{\mathbf{F}}_n^{\star,k+1}(\sigma_{s_n}, \sigma_{s_\star}), \forall n \in \mathcal{N}, s_n \in \mathcal{S}$ from the empirical estimator and the inference.
14: Add a bonus term as $\tilde{\mathbf{F}}_n^{\star,k+1}(\sigma_{s_n}, \sigma_{s_\star}) \leftarrow \hat{\mathbf{F}}_n^{\star,k+1}(\sigma_{s_n}, \sigma_{s_\star}) + \min\left(\delta_{s_n, s_\star}^{\text{inf}}, \sqrt{\frac{1}{\max(2C_{n\star}^k(s_n, s_\star), 1)} \ln\left(\frac{4|\mathcal{S}||\mathcal{A}|N(k-1)H}{\epsilon}\right)}\right)$;

---

Algorithm 3 contains the four key steps in in Section 3.2. In particular, lines 2-9 in Algorithm 3 correspond to **Step 1: Pariwise Comparison**, line 10 denotes the **Step 2: Empirical Preference Estimation**, lines 11-13 are the realization of **Step 3: Preference Inference**, and line 14 is the final **Step 4: Overestimation for Exploration**.

### C.4 Example of Lemma 2

To better understand the importance of Lemma 2, we provide the following example.

**Example 2.** *Assume we have $N = 5$ arms (i.e., 1, 2, 3, 4, 5) and at each time $B = 4$ out of them can be pulled. Without loss of generality, the pulled arms are arm 1, arm 2, arm 3, and arm 4, and their current states are $s_1, s_2, s_3$ and $s_4$. For pairwise comparisons, we can have a total 6 comparisons and get $\hat{\mathbf{F}}_1^2(\sigma_{s_1}, \sigma_{s_2}), \hat{\mathbf{F}}_1^3(\sigma_{s_1}, \sigma_{s_3}), \hat{\mathbf{F}}_1^4(\sigma_{s_1}, \sigma_{s_4}), \hat{\mathbf{F}}_2^3(\sigma_{s_2}, \sigma_{s_3}), \hat{\mathbf{F}}_2^4(\sigma_{s_2}, \sigma_{s_4})$ and $\hat{\mathbf{F}}_3^4(\sigma_{s_3}, \sigma_{s_4})$ with*

$\hat{\mathbf{F}}_m^n(\sigma_{s_m}, \sigma_{s_n})$ *being defined as*

$$\mathbf{F}_m^n(\sigma_{s_m}, \sigma_{s_n}) := \mathbf{F}((m-1)|\mathcal{S}| + \sigma_{s_m}, (n-1)|\mathcal{S}| + \sigma_{s_n}) = \frac{\exp(r_m(s_m))}{\exp(r_m(s_m)) + \exp(r_n(s_n))}.$$

*However, Lemma 2 indicates we only need to do $B - 1 = 3$ comparisons and infer the rest of them. For example, we randomly select the comparison between arm 1 and arm 2, arm 1 and arm 3, arm 1 and arm 4. Hence, we have $\hat{\mathbf{F}}_1^2(\sigma_{s_1}, \sigma_{s_2})$, $\hat{\mathbf{F}}_1^3(\sigma_{s_1}, \sigma_{s_3})$, and $\hat{\mathbf{F}}_1^4(\sigma_{s_1}, \sigma_{s_4})$. Next, we can infer $\hat{\mathbf{F}}_2^3(\sigma_{s_2}, \sigma_{s_3})$, $\hat{\mathbf{F}}_3^4(\sigma_{s_3}, \sigma_{s_4})$, and $\hat{\mathbf{F}}_2^4(\sigma_{s_2}, \sigma_{s_4})$ as follows*

$$\hat{\mathbf{F}}_2^{3,inf}(s_2, s_3) = \frac{(1 - \hat{\mathbf{F}}_1^2(s_1, s_2))\hat{\mathbf{F}}_1^3(s_1, s_3)}{(1 - \hat{\mathbf{F}}_1^2(s_1, s_2))\hat{\mathbf{F}}_1^3(s_1, s_3) + (1 - \hat{\mathbf{F}}_1^3(s_1, s_3))\hat{\mathbf{F}}_1^2(s_1, s_2)},$$

$$\hat{\mathbf{F}}_3^{4,inf}(s_3, s_4) = \frac{(1 - \hat{\mathbf{F}}_1^3(s_1, s_3))\hat{\mathbf{F}}_1^4(s_1, s_4)}{(1 - \hat{\mathbf{F}}_1^3(s_1, s_3))\hat{\mathbf{F}}_1^4(s_1, s_4) + (1 - \hat{\mathbf{F}}_1^4(s_1, s_4))\hat{\mathbf{F}}_1^3(s_1, s_3)},$$

$$\hat{\mathbf{F}}_2^{4,inf}(s_2, s_4) = \frac{(1 - \hat{\mathbf{F}}_1^2(s_1, s_2))\hat{\mathbf{F}}_1^4(s_1, s_4)}{(1 - \hat{\mathbf{F}}_1^2(s_1, s_2))\hat{\mathbf{F}}_1^4(s_1, s_4) + (1 - \hat{\mathbf{F}}_1^4(s_1, s_4))\hat{\mathbf{F}}_1^2(s_1, s_2)}.$$

## D    PROOFS OF LEMMAS IN SECTION 3

In this section, we provide the missing proofs of the lemmas in Section 3.

### D.1    PROOF OF PROPOSITION 1

According to the Bradley-Terry model in (2), we have the preference of arbitrary arm $n$ with arbitrary state $s_n$ over the reference arm $\star$ with reference state $s_\star$ as

$$\mathbf{F}_n^\star(\sigma_s, \sigma_{s_\star}) = \frac{\exp(r_n(s))}{\exp(r_n(s)) + \exp(r_\star(s_\star))}. \tag{18}$$

With standard manipulation, we have

$$\exp(r_\star(s_\star)) = \frac{1 - \mathbf{F}_n^\star(\sigma_s, \sigma_{s_\star})}{\mathbf{F}_n^\star(\sigma_s, \sigma_{s_\star})} \exp(r_n(s)). \tag{19}$$

Taking logarithms to both sides further leads to

$$r_n(s) = r_\star(s_\star) + \ln \frac{\mathbf{F}_n^\star(\sigma_s, \sigma_{s_\star})}{1 - \mathbf{F}_n^\star(\sigma_s, \sigma_{s_\star})}, \tag{20}$$

with the fact that $\mathbf{F}_\star^n(\sigma_{s_\star}, \sigma_s) = 1 - \mathbf{F}_n^\star(\sigma_s, \sigma_{s_\star})$. This completes the proof.

### D.2    PROOF OF PROPOSITION 2

Recall that the underlying reward for each arm and state $r_n(s)$ belongs to the range of $[0, 1]$, we can easily find that any preference value in $\mathbf{F}$ lies in the range of $[\frac{1}{1+e}, \frac{e}{1+e}]$ under the Bradely-Terry model in (2). Defining an auxiliary function $Q(x) = \ln \frac{x}{1-x}$ and taking the derivative of $Q(x)$ with respect to $x$, and thus we have

$$\frac{dQ(x)}{dx} = \frac{d}{dx}\left[\ln \frac{x}{1-x}\right] = \frac{1}{x(1-x)}. \tag{21}$$

Notice that $\frac{dQ(x)}{x} > 0$ for $x \in [\frac{1}{1+e}, \frac{e}{1+e}]$, and hence $Q(x)$ is monotonically increasing with $x$ in this domain. Substituting the endpoints into $Q(x)$, we can easily find that $Q(x) \in [-1, 1]$. Since $\mathbf{F}_n^\star(\sigma_s, \sigma_{s_\star})$ is in the range of $[\frac{1}{1+e}, \frac{e}{1+e}]$, we have $Q_n(s)$ being monotonically increasing function and bounded in $[-1, 1]$. This completes the proof.

### D.3 PROOF OF LEMMA 1

Define the following event that there exists at least one state-action pair $(s, a)$ of any arm $n$ such that the true transition kernel $P_n^k(s'|s, a)$ lies outside of the confidence ball $\mathcal{P}_n^k(s, a)$ defined in (4), i.e.,

$$\mathcal{E}_{p,n}^k := \{\exists(s, a), |P_n(s'|s, a) - \hat{P}_n^k(s'|s, a)| > \delta_n^t(s, a)\}. \tag{22}$$

The probability that the failure event $\mathbb{1}_{\{\mathcal{E}_{p,n}^k\}}$ occur is bounded as follows. We first introduce the Chernoff-Hoeffding inequality (Hoeffding, 1994) in the following lemma.

**Lemma 7** (Hoeffding inequality (Hoeffding, 1994))**.** *Let $X_1, X_2, \ldots$ be independent random variables with $b \leq |X_i| \leq c$ for all $i$. Define $S_n = X_1 + \ldots + X_n$. Then for all $\epsilon > 0$*

$$Pr(S_n - \mathbb{E}[S_n] > \epsilon) \leq \exp\left(-\frac{\epsilon^2}{2n(c-b)^2}\right).$$

Provided the confidence interval $\delta_n^k(s, a) = \sqrt{\frac{1}{2Z_n^{k-1}(s,a)} \ln\left(\frac{4|\mathcal{S}||\mathcal{A}|N(k-1)H}{\epsilon}\right)}$, we have the following inequality according to Lemma 7, $\forall k \geq 2$,

$$\mathbb{P}\big(|P_n(s'|s, a) - \hat{P}_n^k(s'|s, a)| > \delta_n^k(s, a)\big) \leq \frac{2\epsilon}{|\mathcal{S}||\mathcal{A}|N(k-1)H}. \tag{23}$$

Using union bound on all states and actions, we have

$$\mathbb{P}(\mathbb{1}_{\{\mathcal{E}_{p,n}^t\}}) \leq \sum_{n=1}^{N} \sum_{(s,a)} \mathbb{P}\big(|P_n(s'|s, a) - \hat{P}_n^k(s'|s, a)| > \delta_n^t(s, a)\big)$$

$$\leq \frac{2\epsilon}{(k-1)H} \leq 2\epsilon.$$

Since the event that $P_n \in \mathcal{P}_n^k$ is the complementary event of $\mathcal{E}_{p,n}^k$, we have that

$$\mathbb{P}(\mathbb{1}_{\{P_n \in \mathcal{P}_n^k\}}) = 1 - \mathbb{P}(\mathbb{1}_{\{\mathcal{E}_{p,n}^t\}}) \geq 1 - 2\epsilon. \tag{24}$$

This completes the proof.

### D.4 PROOF OF LEMMA 2

According to Bradley-Terry model in (2), we have

$$\mathbf{F}(j, j_1) = \frac{\exp(r_j)}{\exp(r_j) + \exp(r_{j_1})}, \ \mathbf{F}(j, j_2) = \frac{\exp(r_j)}{\exp(r_j) + \exp(r_{j_2})}. \tag{25}$$

Hence, we have

$$\mathbf{F}(j_1, j_2) = \frac{\exp(r_{j_1})}{\exp(r_{j_1}) + \exp(r_{j_2})}$$

$$= \frac{\exp(r_j)(\frac{1-\mathbf{F}(j,j_1)}{\mathbf{F}(j,j_1)})}{\exp(r_i)(\frac{1-\mathbf{F}(j,j_1)}{\mathbf{F}(j,j_1)}) + \exp(r_i)(\frac{1-\mathbf{F}(j,j_2)}{\mathbf{F}(j,j_2)})}$$

$$= \frac{\frac{1-\mathbf{F}(j,j_1)}{\mathbf{F}(j,j_1)}}{\frac{1-\mathbf{F}(j,j_1)}{\mathbf{F}(j,j_1)} + \frac{1-\mathbf{F}(j,j_2)}{\mathbf{F}(j,j_2)}}$$

$$= \frac{(1-\mathbf{F}(j,j_1))\mathbf{F}(j,j_2)}{(1-\mathbf{F}(j,j_1))\mathbf{F}(j,j_2) + (1-\mathbf{F}(j,j_2))\mathbf{F}(j,j_1)}, \tag{26}$$

where the second inequality is due to (25). That being said, we can infer the exact value of $\mathbf{F}(j_1, j_2)$ if we have the exact value of $\mathbf{F}(j, j_1)$ and $\mathbf{F}(j, j_2)$ for arbitrary $j$.

Now assume that we have the empirical estimated values of $\mathbf{F}(j, j_1)$ and $\mathbf{F}(j, j_2)$ satisfying $|\hat{\mathbf{F}}(j, j_1) - \mathbf{F}(j, j_1)| \leq \delta_1$ and $|\hat{\mathbf{F}}(j, j_2) - \mathbf{F}(j, j_2)| \leq \delta_2$ with probability at least $1 - 2\epsilon$, and we have the following estimation

$$\hat{\mathbf{F}}^{\text{inf}}(j_1, j_2) = \frac{(1 - \hat{\mathbf{F}}(j, j_1))\hat{\mathbf{F}}(j, j_2)}{(1 - \hat{\mathbf{F}}(j, j_1))\hat{\mathbf{F}}(j, j_2) + (1 - \hat{\mathbf{F}}(j, j_2))\hat{\mathbf{F}}(j, j_1)}. \tag{27}$$

Our goal is to find the error bound of $|\hat{\mathbf{F}}^{\text{inf}}(j_1, j_2) - \mathbf{F}(j_1, j_2)|$ given $\delta_1$ and $\delta_2$, which can be expressed as

$$|\hat{\mathbf{F}}^{\text{inf}}(j_1, j_2) - \mathbf{F}(j_1, j_2)| = \left| \frac{(1 - \hat{\mathbf{F}}(j, j_1))\hat{\mathbf{F}}(j, j_2)}{(1 - \hat{\mathbf{F}}(j, j_1))\hat{\mathbf{F}}(j, j_2) + (1 - \hat{\mathbf{F}}(j, j_2))\hat{\mathbf{F}}(j, j_1)} \right.$$
$$\left. - \frac{(1 - \mathbf{F}(j, j_1))\mathbf{F}(j, j_2)}{(1 - \mathbf{F}(j, j_1))\mathbf{F}(j, j_2) + (1 - \mathbf{F}(j, j_2))\mathbf{F}(j, j_1)} \right|. \tag{28}$$

To bound (28), we define the auxiliary function $f(x, y) := \frac{(1-x)y}{x+y-2xy}$ and show that $f(x, y)$ is Lipschitz continuous in the following lemma.

**Lemma 8.** *Provide $f(x, y) = \frac{(1-x)y}{x+y-2xy}$, and have $x, y \in \left[\frac{1}{1+e}, \frac{e}{1+e}\right]$, $f(x, y)$ is Lipschitz continuous, with a valid lipschitz constant $L = 1.3$.*

*Proof.* To determine whether $f(x, y)$ is Lipschitz continuous over the interval $\left[\frac{1}{1+e}, \frac{e}{1+e}\right]$, we need to analyze the partial derivatives of $f(x, y)$ and check if they are bounded within this interval. Let us start by calculating the partial derivatives of $f$ with respect to $x$ and $y$

$$\frac{df(x, y)}{dx} = \frac{y^2 - y}{(x + y - 2xy)^2}, \qquad \frac{df(x, y)}{dy} = \frac{x - x^2}{(x + y - 2xy)^2}. \tag{29}$$

In the given interval of $x$ and $y$, the partial derivatives are bounded because both $x$ and $y$ are within a closed interval away from 0 and 1, and thus function $f(x, y)$ is Lipschitz continuous, i.e., we have

$$|f(x_1, y_1) - f(x_2, y_2)| \leq L(|x_1 - x_2| + |y_1 - y_2|), \tag{30}$$

with $L$ being the Lipschitz constant. To find the Lipschitz constant $L$, we need to find the maximum absolute values of these partial derivatives in the given domain $x, y \in \left[\frac{1}{1+e}, \frac{e}{1+e}\right]$. Therefore, we evaluate these expressions numerically or analytically at the boundary points to estimate $L$. The maximum values of $|f_x(x, y)|$ and $|f_y(x, y)|$ will be determined within these bounds. Finally,

$$L = \max\{|f_x(x, y)|, |f_y(x, y)|\} \text{ for } x, y \in \left[\frac{1}{1+e}, \frac{e}{1+e}\right]. \tag{31}$$

After numerical evaluation, we can infer that $L \leq 1.3$ and it is achieved by $\frac{df(x,y)}{dy}$ at $x = y = \frac{1}{1+e}$, which completes the proof. $\square$

According to Lemma 8, we have the following inequality

$$|\hat{\mathbf{F}}^{\text{inf}}(j_1, j_2) - \mathbf{F}(j_1, j_2)| = \left| f(\hat{\mathbf{F}}(j, j_1), \hat{\mathbf{F}}(j, j_2)) - f(\mathbf{F}(j, j_1), \mathbf{F}(j, j_2)) \right|$$

$$= \left| f(\hat{\mathbf{F}}(j, j_1), \hat{\mathbf{F}}(j, j_2)) - f(\mathbf{F}(j, j_1), \hat{\mathbf{F}}(j, j_2)) \right.$$
$$\left. + f(\mathbf{F}(j, j_1), \hat{\mathbf{F}}(j, j_2)) - f(\mathbf{F}(j, j_1), \mathbf{F}(j, j_2)) \right|$$

$$\overset{(a)}{\leq} \left| f(\hat{\mathbf{F}}(j, j_1), \hat{\mathbf{F}}(j, j_2)) - f(\mathbf{F}(j, j_1), \hat{\mathbf{F}}(j, j_2)) \right|$$

$$+ \left| f(\mathbf{F}(j, j_1), \hat{\mathbf{F}}(j, j_2)) - f(\mathbf{F}(j, j_1), \mathbf{F}(j, j_2)) \right|$$

$$\overset{(b)}{\leq} L(\delta_1 + \delta_2), \tag{32}$$

where $(a)$ is due to triangular inequality and $(b)$ comes from the Lipschitz continuity of function $f$ in Lemma 8.

### D.5 PROOF OF LEMMA 3

According to Proposition 1, any reward $r_n(s)$ can be equivalently denoted as $r_\star(s_\star) + Q_n(s)$. Then we can rewrite the LP in (6) as

$$\max_{\mu^\pi} \sum_{n=1}^{N} \sum_{(s,a) \in \mathcal{S} \times \mathcal{A}} \mu_n(s,a)(r_*(s_*) + Q_n(s))$$

$$\text{s.t.} \sum_{n=1}^{N} \sum_{(s,a) \in \mathcal{S} \times \mathcal{A}} a\mu_n(s,a) \leq B,$$

$$\sum_{a} \mu_n(s,a) = \sum_{s'} \sum_{a'} \mu_n(s',a') P_n(s',a',s), \quad \forall n \in \mathcal{N},$$

$$\sum_{(s,a) \in \mathcal{S} \times \mathcal{A}} \mu_n(s,a) = 1, \quad \mu_n(s,a) \geq 0, \quad \forall n \in \mathcal{N}, s \in \mathcal{S}, a \in \mathcal{A}. \tag{33}$$

Hence, the objective can be expressed as

$$\max_{\mu^\pi} \left[ \sum_{n=1}^{N} \sum_{(s,a) \in \mathcal{S} \times \mathcal{A}} \mu_n(s,a) Q_n(s) + \sum_{n=1}^{N} \sum_{(s,a) \in \mathcal{S} \times \mathcal{A}} \mu_n(s,a) r_\star(s_\star) \right], \tag{34}$$

which equals to

$$\max_{\mu^\pi} \left[ \sum_{n=1}^{N} \sum_{(s,a) \in \mathcal{S} \times \mathcal{A}} \mu_n(s,a) Q_n(s) \right] + N r_\star(s_\star), \tag{35}$$

due to the fact that occupancy measure is a probability measure such that $\sum_{(s,a) \in \mathcal{S} \times \mathcal{A}} \mu_n(s,a) = 1$. Therefore, the optimal solution of the LP in (33) is equivalent to

$$\max_{\mu_\pi} \sum_{n=1}^{N} \sum_{(s,a) \in \mathcal{S} \times \mathcal{A}} \mu_n(s,a) Q_n(s)$$

$$\text{s.t. constraints in (7)-(9).}$$

This completes the proof.

## E PROOF OF THEOREM 1

In this section, we provide detailed proof for the main result in Theorem 1. We first prove that the regret decomposition as shown in Lemma 4, and then separately characterize the regret of $Term_1$ in Lemma 5 and the regret of $Term_2$ in Lemma 6.

### E.1 PROOF OF LEMMA 4

According to the definition of regret in (3), We can decompose the regret $Reg(\{\pi^k\}_{k=1}^K, T)$ as follows:

$$Reg(\{\pi^k\}_{k=1}^K, T) = J(\pi^{opt}, T) - J(\{\pi^k\}_{k=1}^K, T)$$

$$= \underbrace{J(\pi^{opt}, T) - J(\{\tilde{\pi}^k, \forall k\}, \{\tilde{\mathbf{F}}^k, \forall k\}, T)}_{Term_0 \leq 0}$$

$$+ \underbrace{J(\{\tilde{\pi}^k, \forall k\}, \{\tilde{\mathbf{F}}^k, \forall k\}, T) - J(\{\tilde{\pi}^k, \forall k\}, \mathbf{F}, T)}_{Term_1}$$

$$+ \underbrace{J(\{\tilde{\pi}^k, \forall k\}, \mathbf{F}, T) - J(\{\pi^k, \forall k\}, \mathbf{F})}_{Term_2}$$

$$\leq \underbrace{J(\{\tilde{\pi}^k, \forall k\}, \{\tilde{\mathbf{F}}^k, \forall k\}, T) - J(\{\tilde{\pi}^k, \forall k\}, \mathbf{F}, T)}_{Term_1}$$

$$+ \underbrace{J(\{\tilde{\pi}^k, \forall k\}, \mathbf{F}, T) - J(\{\pi^k, \forall k\}, \mathbf{F})}_{Term_2}. \tag{36}$$

The key for (36) to hold is to guarantee that $Term_0 \leq 0$, which is shown in the following lemma.

**Lemma 9** (**Optimism**). *There exists a set of occupancy measures $\mu_{\pi^*} := \{\mu_n^*(s, a), \forall n \in \mathcal{N}, s \in \mathcal{S}, a \in \mathcal{A}\}$ under a policy $\pi^*$ that optimally solve the LP in (6)-(9). In addition, $J(\pi^*)$ for the LP in (6)-(9) is no less than $J(\pi^{opt})$ for the original RMAB problem in (1).*

*Proof.* It is straightforward to show that the LP in (6)-(9) is equivalent to the original RMAB problem in (1) if the "hard" activation constraint in (1) is relaxed to an averaged one as $\liminf_{T \to \infty} \frac{1}{T} \mathbb{E}_\pi \left[ \sum_{t=1}^T \sum_{n=1}^N A_n^t \right] \leq B$. To prove Lemme 9, it is sufficient to show that the relaxed problem achieves no less average reward than the original problem in (1). The proof is straightforward since the relaxed constraint expands the feasible region of the original problem in (1). Denote the feasible region of the original problem as

$$\Delta := \left\{ A_n^t, \forall t \left| \sum_{n=1}^N A_n^t \leq B, \forall t \right. \right\}, \tag{37}$$

and the feasible region of the relaxed constraint as

$$\Delta' := \left\{ A_n^t, \forall t \left| \liminf_{T \to \infty} \frac{1}{T} \mathbb{E}_\pi \left[ \sum_{t=1}^T \sum_{n=1}^N A_n^t \leq B \right] \right. \right\}. \tag{38}$$

It is clear that the relaxed constraint expands the feasible region of the original problem in (1), i.e., $\Delta \subseteq \Delta'$. Therefore, the relaxed problem (i.e., the LP in (6)-(9)) achieves an objective value no less than that of the original problem in (1) because the original optimal solution is also inside the relaxed feasibility set (Altman, 1999), i.e., $\Delta'$. Denote the optimal occupancy measures of LP in (6)-(9) as $\mu_{\pi^*} := \{\mu_n^*(s, a), \forall n \in \mathcal{N}, s \in \mathcal{S}, a \in \mathcal{A}\}$ under a stationary policy $\pi^*$ induced by $\{\mu_n^*(s, a), \forall n \in \mathcal{N}, s \in \mathcal{S}, a \in \mathcal{A}\}$, and hence the maximum average reward achieved for the LP in (6)-(9) is equal to $J(\pi^*)$. Therefore, it follows that $J(\pi^*) \geq J(\pi^{opt})$, which completes the proof. $\square$

Hence, $Term_0 \leq 0$ directly follows Lemma 9.

### E.2 PROOF OF LEMMA 5

Since DOPL designs index not directly based on the preference matrix $\mathbf{F}$, but on the preference-reference term $Q_n(s)$ as defined in Proposition 2. To characterize the impact on the preference-reference term $Q_n(s)$ made by the estimation error of $F$, we first define a general function $q(x) := \ln \frac{x}{1-x}$. We aim to analyze the impact of a small perturbation in $x$, denoted by $\varepsilon$, on $q(x)$ with the aid of *Taylor Series Approximation* of $q(x)$. This is presented in the following lemma.

**Lemma 10.** *Assume $\hat{x} = x + \varepsilon$ where $\varepsilon$ is small value in $(0, 1)$. Then we have*

$$|q(\hat{x}) - q(x)| \leq \frac{\varepsilon}{x(1-x)}.$$

*Proof.* We can approximate $q(\hat{x})$ using a first-order Taylor expansion around $x$:

$$q(\hat{x}) \approx q(x) + \frac{dq(x)}{dx} \cdot (\hat{x} - x). \tag{39}$$

Following the chain rule, we have the derivative as follows

$$\frac{dq(x)}{dx} = \frac{1-x}{x} \cdot \frac{d}{dx}\left(\frac{x}{1-x}\right) = \frac{1-x}{x} \cdot \frac{1}{(1-x)^2} = \frac{1}{x(1-x)}. \tag{40}$$

Using this derivative, the approximation becomes:

$$q(\hat{x}) \approx q(x) + \frac{\varepsilon}{x(1-x)}. \tag{41}$$

Substitute this into the Taylor expansion:

$$|q(\hat{x}) - q(x)| \leq \frac{\varepsilon}{x(1-x)}. \tag{42}$$

This completes the proof. $\square$

We need to characterize the performance guarantee of Algorithm 3, which is used for the regret analysis for DOPL. Provided Lemma 7, we have the following result w.r.t the estimation error of preference matrix $\mathbf{F}$. Hoeffding's Inequality in Lemma 7 states that for independent and bounded random variables $X_1, X_2, \ldots, X_n$ the following holds with probability at least $1 - \epsilon$:

$$Pr\left(\left|\frac{1}{n}\sum_{i=1}^{n} X_i - \mathbb{E}[X]\right| \geq d\right) \leq 2\exp\left(\frac{-2nd^2}{(c-b)^2}\right). \tag{43}$$

Applying this to our Bernoulli trials where $b = 0$ and $c = 1$:

$$\mathbb{P}\left(\left|\hat{\mathbf{F}}_i^{j,k}(\sigma_{s_i}, \sigma_{s_j}) - \mathbf{F}_i^j(\sigma_{s_i}, \sigma_{s_j})\right| \geq d\right) \leq 2\exp\left(-2C_{ij}^k(s_i, s_j)d^2\right). \tag{44}$$

We focus on one specific column of the preference matrix $\mathbf{F}$, i.e., $\mathbf{F}(:, (\star - 1)|\mathcal{S}| + \sigma_{s_\star})$. Setting $d = \min\left(\delta_{s_n, s_\star}^{\inf}, \sqrt{\frac{1}{\max(2C_{n_\star}^k(s_n, s_\star), 1)}\ln\left(\frac{4|\mathcal{S}||\mathcal{A}|N(k-1)H}{\epsilon}\right)}\right)$. Specifically, when $d = \sqrt{\frac{1}{\max(2C_{n_\star}^k(s_n, s_\star), 1)}\ln\left(\frac{4|\mathcal{S}||\mathcal{A}|N(k-1)H}{\epsilon}\right)}$, we have :

$$\mathbb{P}\left(\left|\hat{\mathbf{F}}_n^{\star,k}(s_n, s_\star) - \mathbf{F}_n^\star(s_n, s_\star)\right| \geq d\right) \leq 2\epsilon. \tag{45}$$

When $d = \delta_{s_n, s_\star}^{\inf}$, we have the inequality in (45) again according to Lemma 2.

Next, we begin to bound $Term_1$. For the ease of expression, we define $\omega_n^k$ as the vector containing all $\omega_n^k(s, a), \forall s \in \mathcal{S}, a \in \mathcal{A}$, and $Q_n(\mathbf{F})$ as the vector containing all $Q_n(s), \forall s \in \mathcal{S}$. Then, according to the definition, we can rewrite $Term_1$ as

$$\begin{aligned}
Term_1 &= J(\{\tilde{\pi}^k, \forall k\}, \{\tilde{\mathbf{F}}^k, \forall k\}, T) - J(\{\tilde{\pi}^k, \forall k\}, \mathbf{F}, T) \\
&= H \cdot \sum_{k=1}^{K}\sum_{n=1}^{N}\langle\omega_n^k, Q_n(\tilde{\mathbf{F}}^k) - Q_n(\mathbf{F})\rangle \\
&= H \cdot \sum_{k=1}^{K}\sum_{n=1}^{N}\langle\omega_n^k, Q_n(\tilde{\mathbf{F}}^k) - Q_n(\hat{\mathbf{F}}^k) + Q_n(\hat{\mathbf{F}}^k) - Q_n(\mathbf{F})\rangle \\
&= H \cdot \sum_{k=1}^{K}\sum_{n=1}^{N}\langle\omega_n^k, Q_n(\tilde{\mathbf{F}}^k) - Q_n(\hat{\mathbf{F}}^k)\rangle + H \cdot \sum_{k=1}^{K}\sum_{n=1}^{N}\langle\omega_n^k, Q_n(\hat{\mathbf{F}}^k) - Q_n(\mathbf{F})\rangle, \tag{46}
\end{aligned}$$

where the second equality is due to the definition of $\tilde{\pi}^k$ and Lemma 3.

According to Lemma 10, we have

$$Term_1 \leq \max_{i,j} \frac{H}{\mathbf{F}(i,j)(1-\mathbf{F}(i,j))} \left( \sum_{k=1}^{K} \sum_{n=1}^{N} \langle \omega_n^k, \tilde{\mathbf{F}}^k(:, (\star-1)|\mathcal{S}|+\sigma_{s_\star}) - \hat{\mathbf{F}}^k(:, (\star-1)|\mathcal{S}|+\sigma_{s_\star}) \rangle \right.$$

$$\left. + \sum_{k=1}^{K} \sum_{n=1}^{N} \langle \omega_n^k, \hat{\mathbf{F}}^k(:, (\star-1)|\mathcal{S}|+\sigma_{s_\star}) - \mathbf{F}(:, (\star-1)|\mathcal{S}|+\sigma_{s_\star}) \rangle \right). \qquad (47)$$

According to definition of $\tilde{\mathbf{F}}^k$ and $\hat{\mathbf{F}}^k$, we have at least probability $1-2\epsilon$ based on (45) that

$$Term_1 \leq \max_{i,j} \frac{2}{\mathbf{F}(i,j)(1-\mathbf{F}(i,j))} \sum_{k=1}^{K} \sum_{n=1}^{N} \langle H \cdot \omega_n^k, \mathbf{d}_n^k \rangle, \qquad (48)$$

with $\mathbf{d}_n^k \in \mathbb{R}^{|\mathcal{S}| \times 1}$ being the vector containing all $d_n^k(s_n)$, $\forall s_n \in \mathcal{S}$, with $d_n^k(s_n) = \min\left( \delta_{s_n,s_\star}^{\inf}, \sqrt{\frac{1}{\max(2C_{n\star}^k(s_n,s_\star),1)} \ln\left( \frac{4|\mathcal{S}||\mathcal{A}|N(k-1)H}{\epsilon} \right)} \right)$.

Next, we bound $\sum_{k=1}^{K} \sum_{n=1}^{N} \langle H \cdot \omega_n^k, \mathbf{d}_n^k \rangle$ in the following lemma.

**Lemma 11.** *The following inequality holds*

$$\sum_{k=1}^{K} \sum_{n=1}^{N} \langle H \cdot \omega_n^k, \mathbf{d}_n^k \rangle \leq \sqrt{2|\mathcal{S}|} \sqrt{\ln \frac{4|\mathcal{S}||\mathcal{A}|NT}{\epsilon}} \cdot \sqrt{NT}.$$

*Proof.* Since $d_n^k(s_n) = \min\left( \delta_{s_n,s_\star}^{\inf}, \sqrt{\frac{1}{\max(2C_{n\star}^k(s_n,s_\star),1)} \ln\left( \frac{4|\mathcal{S}||\mathcal{A}|N(k-1)H}{\epsilon} \right)} \right)$, we have

$$\sum_{k=1}^{K} \sum_{n=1}^{N} \langle H \cdot \omega_n^k, \mathbf{d}_n^k \rangle \leq \sum_{k=1}^{K} \sum_{n=1}^{N} \left\langle H \cdot \omega_n^k, \left[ \sqrt{\frac{1}{\max(2C_{n\star}^k(s_i,s_\star),1)} \ln\left( \frac{4|\mathcal{S}||\mathcal{A}|N(k-1)H}{\epsilon} \right)} \right]_{i=1}^{|\mathcal{S}|} \right\rangle. \qquad (49)$$

Hence, the proof goes as follows.

$$\sum_{k=1}^{K} \sum_{n=1}^{N} \left\langle H \cdot \omega_n^k, \left[ \sqrt{\frac{1}{\max(2C_{n\star}^k(s_i,s_\star),1)} \ln\left( \frac{4|\mathcal{S}||\mathcal{A}|N(k-1)H}{\epsilon} \right)} \right]_{i=1}^{|\mathcal{S}|} \right\rangle$$

$$= \sum_{k=1}^{K} \sum_{n=1}^{N} \sum_{s,a} \left\langle H \cdot \omega_n^k(s,a), \sqrt{\frac{1}{\max(2C_{n\star}^k(s,s_\star),1)} \ln\left( \frac{4|\mathcal{S}||\mathcal{A}|N(k-1)H}{\epsilon} \right)} \right\rangle$$

$$\overset{(a)}{\leq} \sqrt{\ln \frac{4|\mathcal{S}||\mathcal{A}|NT}{\epsilon}} \sum_{t=1}^{T} \sum_{n=1}^{N} \sum_{(s,a)} \mathbb{1}(s_n(t)=s, a_n(t)=a) \sqrt{\frac{1}{2Z_n^t(s,a)}}$$

$$= \sqrt{\ln \frac{4|\mathcal{S}||\mathcal{A}|NT}{\epsilon}} \sum_{t=1}^{T} \sum_{n=1}^{N} \sum_{(s,a)} \frac{\mathbb{1}(s_n(t)=s, a_n(t)=a)}{\sqrt{2Z_n^t(s,a)}}$$

$$\overset{(b)}{\leq} 2\sqrt{\ln \frac{4|\mathcal{S}||\mathcal{A}|NT}{\epsilon}} \sum_{n=1}^{N} \sum_{(s,a)} \sqrt{Z_n^T(s,a)}$$

$$\overset{(c)}{\leq} 2\sqrt{\ln \frac{4|\mathcal{S}||\mathcal{A}|NT}{\epsilon}} \sum_{n=1}^{N} |\mathcal{S}||\mathcal{A}| \sqrt{\frac{\sum_{(s,a)} Z_n^T(s,a)}{|\mathcal{S}||\mathcal{A}|}} \quad \text{Jensen's inequality}$$

$$\overset{(d)}{\leq} 2\sqrt{\ln \frac{4|\mathcal{S}||\mathcal{A}|NT}{\epsilon}} \sum_{n=1}^{N} \sqrt{|\mathcal{S}||\mathcal{A}|T} \quad \text{due to} \sum_{s,a} Z_n^T(s,a) \leq T$$

$$= 2N\sqrt{\ln \frac{4|\mathcal{S}||\mathcal{A}|NT}{\epsilon}} \sqrt{|\mathcal{S}||\mathcal{A}|T}$$

$$\leq 2\sqrt{2}\sqrt{\ln\frac{4|\mathcal{S}||\mathcal{A}|NT}{\epsilon}}\sqrt{N^2|\mathcal{S}|T} \quad \text{due to}|\mathcal{A}| = 2 \tag{50}$$

where (a) follows since $\omega_n^k$ is a probability measure, $H \cdot \omega_n^k(s, a) = Z_n^k(s, a) - Z_n^{k-1}(s, a) = \sum_{h=1}^H \mathbb{1}(s_n(k, h) = s, a_n(k, h) = a)$, (b) is due to the fact that for any sequence of numbers $w_1, w_2, ..., w_T$ with $0 \leq w_k$, and define $W_k := \sum_{i=1}^k w_i$, we have

$$\sum_{k=1}^T \frac{w_k}{\sqrt{W_k}} \leq 2\sqrt{2}\sqrt{W_T}.$$

(c) follows Jensen's inequality and the fact that $\sum_{s,a} Z_n^T(s, a) = T$, and (d) uses the fact that $|\mathcal{A}| = 2$. This completes the proof. $\qquad\square$

**Bound on $Term_1$.** Combining the results in (48) and Lemma 11, we can bound $Term_1$ as

$$Term_1 \leq \max_{i,j} \frac{2}{\mathbf{F}(i, j)(1 - \mathbf{F}(i, j))}\sqrt{2NT|\mathcal{S}|\ln\frac{4|\mathcal{S}||\mathcal{A}|NT}{\epsilon}}.$$

### E.3 PROOF OF LEMMA 6

To characterize the regret caused by $Term_2$, we leverage similar techniques from Xiong et al. (2022b). We first introduce the definition of ergodicity coefficient as shown Arapostathis et al. (1993); Akbarzadeh & Mahajan (2022), which depicts the transition property of the whole $N$ arms system. Denote the global state for the whole $N$-arm system as $\mathbf{s} \in \mathcal{S}^N := [s_1, s_2, \ldots, s_N]$, the corresponding actions under the index policy $\pi$ as $\pi(\mathbf{s})$, and the unknown MDPs as $\Lambda := [\lambda_1, \lambda_2, \ldots, \lambda_N]$ with $\lambda_n := \{P_n, \mathbf{F}_n^\star\}$. Hence, we have a transition kernel for the global system as $P_\Lambda(\cdot|\mathbf{s}, \pi(\mathbf{s})), \forall \mathbf{s} \in \mathcal{S}^N$. The ergodicity coefficient is then defined as follows.

**Definition 1** (**Ergodicity coefficient**). *The ergodicity coefficient of a system with transition kernel $P_\Lambda$ is defined*

$$D_{P_\Lambda} := 1 - \min_{\mathbf{s},\mathbf{s}'} \sum_{\mathbf{z} \in \mathcal{S}^N} \min\{P_\Lambda(\mathbf{z}|\mathbf{s}, \pi^\star(\mathbf{s})), P_\Lambda(\mathbf{z}|\mathbf{s}', \pi^\star(\mathbf{s}'))\}, \tag{51}$$

*and $D := \sup_\Lambda D_{P_\Theta}$ as the maximum value.*

Since the dynamics of the arms are independent, the definition of contraction factor implies that a sufficient condition is that for every arm, and every pair of state-action pairs, there exists a next state that can be reached from both state-action pairs with positive probability in one step. Let $\pi_\Lambda$ denote the proposed index policy corresponding to the transition model $\Lambda$ and $P_\Lambda$ be the controlled transition matrix under policy $\pi_\Lambda$. Denote $J_\Lambda$ as the average reward of policy $\pi_\Lambda$ and $R_\Lambda$ does not depend on the initial state and satisfy the average reward Bellman equation (Altman, 1999; Puterman, 1994),

$$J_\Lambda + F_\Lambda(\mathbf{s}) = R(\mathbf{s}, \pi_\Lambda(\mathbf{s})) + [P_\Lambda F_\Lambda](s), \quad \forall \mathbf{s} \in \mathcal{S}^N, \tag{52}$$

where $F_\Lambda(\mathbf{s})$ is the relative value function.

To bound $Term_2$, we first bound the episodic regret $J(\tilde{\pi}^k, \mathbf{F}, H) - J(\pi^k, \mathbf{F}, H), \forall k$.

**Lemma 12.** *With probability at least $1 - 2\epsilon$, the regret for $J(\tilde{\pi}^k, \mathbf{F}, H) - J(\pi^k, \mathbf{F}, H)$ can be expressed as*

$$J(\tilde{\pi}^k, \mathbf{F}, H) - J(\pi^k, \mathbf{F}, H) \leq \left[\sum_{h=1}^H [P_{\tilde{\Lambda}} F_{\tilde{\Lambda}}](\mathbf{s}_h) - F_{\tilde{\Lambda}}(\mathbf{s}_{h+1})\right] + \frac{2B}{1 - D},$$

*where $\tilde{\Lambda}$ is the parameter of the optimistic MDP $\{\tilde{P}_n, \forall n \in \mathcal{N}\}$.*

*Proof.* The proof goes as follows:

$$J(\tilde{\pi}^k, \mathbf{F}, H) - J(\pi^k, \mathbf{F}, H) = H\tilde{J}_\Lambda - \sum_{h=1}^H R(\mathbf{s}_h, \pi^k(\mathbf{s}_h))$$

$$\overset{(a)}{=} \sum_{h=1}^H R(\mathbf{s}_h, \tilde{\pi}^k(\mathbf{s}_h)) + \sum_{h=1}^H [P_{\tilde{\Lambda}} F_{\tilde{\Lambda}}](\mathbf{s}_h) - F_{\tilde{\Lambda}}(\mathbf{s}_h) - \sum_{h=1}^H R(\mathbf{s}(h), \pi^k(\mathbf{s}_h))$$

$$\overset{(b)}{=} \sum_{h=1}^{H} [P_{\tilde{\Lambda}} F_{\tilde{\Lambda}}](\mathbf{s}_h) - F_{\tilde{\Lambda}}(\mathbf{s}_h)$$

$$= \sum_{h=1}^{H} [P_{\tilde{\Lambda}} F_{\tilde{\Lambda}}](\mathbf{s}_h) - F_{\tilde{\Lambda}}(\mathbf{s}_{h+1}) + F_{\tilde{\Lambda}}(\mathbf{s}_{h+1}) - F_{\tilde{\Lambda}}(\mathbf{s}_h)$$

$$= \sum_{h=1}^{H} [P_{\tilde{\Lambda}} F_{\tilde{\Lambda}}](\mathbf{s}_h) - F_{\tilde{\Lambda}}(\mathbf{s}_{h+1}) + F_{\tilde{\Lambda}}(\mathbf{s}_{H+1}) - F_{\tilde{\Lambda}}(\mathbf{s}_1)$$

$$\overset{(c)}{\leq} \sum_{h=1}^{H} [P_{\tilde{\Lambda}} F_{\tilde{\Lambda}}](\mathbf{s}_h) - F_{\tilde{\Lambda}}(\mathbf{s}_{t+1}) + \frac{2B}{1-D}.$$

The equality in (a) directly follows the Bellman equation in (52). (b) is due to the fact that $\sum_{h=1}^{H} R(\mathbf{s}_h, \tilde{\pi}^k(\mathbf{s}_h)) = \sum_{h=1}^{H} R(\mathbf{s}_h, \pi^k(\mathbf{s}_h))$, and (c) follows from Lemma 8 in Xiong et al. (2022b). □

Then, we bound $J(\{\tilde{\pi}^k, \forall k\}, \mathbf{F}, T) - J(\{\pi^k, \forall k\}, \mathbf{F}, T)$ as follows.

**Lemma 13.** *With probability at least $1 - 2\epsilon$, we have the regret bounded conditioned on the good event that the $\tilde{P}_n^k \in \mathcal{P}_n^k$, as*

$$J(\{\tilde{\pi}^k, \forall k\}, \mathbf{F}, T) - J(\{\pi^k, \forall k\}, \mathbf{F}, T) \leq \frac{2\sqrt{2}B}{1-D} \sqrt{\log \frac{4|\mathcal{S}||\mathcal{A}|NT}{\eta}} \cdot \sqrt{NT}.$$

*Proof.* From Lemma 12, we can rewrite the summation over $J(\tilde{\pi}^k, \mathbf{F}, H) - J(\pi^k, \mathbf{F}, H), \forall k$ as follows:

$$J(\{\tilde{\pi}^k, \forall k\}, \mathbf{F}, T) - J(\{\pi^k, \forall k\}, \mathbf{F}, T) - \frac{2BK}{1-D}$$

$$\leq \sum_{k=1}^{K} \sum_{h=1}^{H} [P_{\tilde{\Lambda}^k} F_{\tilde{\Lambda}^k}](\mathbf{s}_h) - F_{\tilde{\Lambda}^k}(\mathbf{s}_{h+1})$$

$$\overset{(a)}{\leq} \sum_{k=1}^{K} \sum_{h=1}^{H} \left| [P_{\tilde{\Lambda}^k} F_{\tilde{\Lambda}^k}](\mathbf{s}_h) - F_{\Lambda^k}(\mathbf{s}_{h+1}) \right|$$

$$\overset{(b)}{\leq} \sum_{k=1}^{K} \sum_{h=1}^{H} \frac{1}{2} \text{span}(F_{\tilde{\Lambda}^k}) \left\| P_{\tilde{\Lambda}^k}(\cdot|\mathbf{s}_h, \mathbf{a}_h) - P_{\Lambda^k}(\cdot|\mathbf{s}_h, \mathbf{a}_h) \right\|_1$$

$$\overset{(c)}{\leq} \frac{B}{1-D} \sum_{k=1}^{K} \sum_{h=1}^{H} \sum_{n=1}^{N} 2\delta_n^k(s, a)$$

$$\leq \frac{2B}{1-D} \sum_{t=1}^{T} \sum_{n=1}^{N} \sqrt{\frac{1}{2Z_n^t(s,a)} \log \frac{4|\mathcal{S}||\mathcal{A}|NT}{\eta}}$$

$$\leq \frac{2B}{1-D} \sqrt{\log \frac{4|\mathcal{S}||\mathcal{A}|NT}{\eta}} \sum_{t=1}^{T} \sum_{n=1}^{N} \sum_{(s',a')} \mathbb{1}(s_n(t) = s', a') \sqrt{\frac{1}{2Z_n^t(s', a')}}$$

$$\overset{(d)}{\leq} \frac{4\sqrt{2}B}{1-D} \sqrt{\log \frac{4|\mathcal{S}||\mathcal{A}|NT}{\eta}} \cdot \sqrt{N^2|\mathcal{S}|T},$$

where (a) follows since $\mathbb{E}\left[ \sum_{k=1}^{K} \sum_{h=1}^{H} [P_{\tilde{\Lambda}^k} F_{\tilde{\Lambda}^k}](\mathbf{s}_h) - F_{\tilde{\Lambda}^k}(\mathbf{s}_{h+1}) \right] = 0$, (b) follows standard linear algebra manipulation (Akbarzadeh & Mahajan, 2022), (c) follows the same reason as (50). □

# F EXPERIMENT DETAILS

The code for these experiments is available at this link. Some of the experiments presented in this paper were run on an M1 Macbook Air and some on a compute cluster with Dual AMD EPYC 7443 with 48 cores and 256GB RAM.

## F.1 APP MARKETING ENVIRONMENT

### F.1.1 OVERVIEW

For a verbose explanation of the setup refer to Appendix B.1. Each user is modeled as a restless arm with 4 possible states. Our simulation was carried out with 10 arms in the system and an arm pull budget of 4 arms per timestep.

### F.1.2 SYSTEM DYNAMICS

**Transition probabilities**:

The transition dynamics are such that without marketing intervention, users tend to decrease in engagement over time. Even the most engaged users only have a 50% chance of staying as engaged. On the other hand, if intervened upon, there's a high probability that the user increases in engagement by two levels.

- **Without intervention:**

$$P_{\text{no\_int}} = \begin{bmatrix} 0.7 & 0.1 & 0.1 & 0.1 \\ 0.5 & 0.3 & 0.1 & 0.1 \\ 0.2 & 0.4 & 0.3 & 0.1 \\ 0.1 & 0.2 & 0.2 & 0.5 \end{bmatrix}$$

- **With intervention:**

$$P_{\text{int}} = \begin{bmatrix} 0.1 & 0.1 & 0.7 & 0.1 \\ 0.1 & 0.1 & 0.1 & 0.7 \\ 0.1 & 0.1 & 0.1 & 0.7 \\ 0.05 & 0.05 & 0.05 & 0.85 \end{bmatrix}$$

**Rewards**:

The latent reward is structured as follows. Higher the engagement level higher the reward.

$$R = \begin{bmatrix} 0 \\ 0.33 \\ 0.66 \\ 1 \end{bmatrix}$$

### F.1.3 HYPERPARAMETERS

Below we detail the hyperparameters used during the training.

| Hyperparameter | Value |
|---|---|
| K (Number of Epochs) | 4000 |
| H (Horizon) | 100 |
| $\epsilon$ (Epsilon) | $1 \times 10^{-5}$ |

Table 1: Hyperparameters used in App Marketing Environment Training.

## F.2 CPAP ENVIRONMENT

### F.2.1 OVERVIEW

For a detailed description of the environment refer Appendix B.2. The environment models each patient undergoing sleep apnea treatment as a restless arm. The patient can be in three possible states based on his level of adherence to the treatment.

Our simulations consist of 20 patients (arms) only 8 of whom can be intervened with by the doctor at any given time (arm budget).

### F.2.2 SYSTEM DYNAMICS

**Transition probabilities**:

Patients can be classified into two clusters based on their transition behaviours. (i) *General patients* and (ii)*High-risk patients*

**General Patients (Arm type 1)**   These patients are very responsive to intervention and also adhere to their treatment fairly well even without intervention as seen in the transition probabilities below.

- **Without intervention:**

$$P_{\text{no\_int}} = \begin{bmatrix} 0.1385 & 0.1 & 0.7615 \\ 0.1 & 0.1 & 0.8 \\ 0.1257 & 0.1245 & 0.7498 \end{bmatrix}$$

- **With intervention:**

$$P_{\text{int}} = \begin{bmatrix} 0.1 & 0.1 & 0.8 \\ 0.1 & 0.1 & 0.8 \\ 0.1 & 0.1 & 0.8 \end{bmatrix}$$

**High-Risk Patients (Arm type 2)**   These patients need constant intervention in order for them to adhere to their treatments.

- **Without intervention:**

$$P_{\text{no\_int}} = \begin{bmatrix} 0.7427 & 0.0741 & 0.1832 \\ 0.3399 & 0.1634 & 0.4967 \\ 0.2323 & 0.1020 & 0.6657 \end{bmatrix}$$

- **With intervention:**

$$P_{\text{int}} = \begin{bmatrix} 0.1427 & 0.3741 & 0.4832 \\ 0.1399 & 0.1 & 0.7601 \\ 0.1323 & 0.1 & 0.7677 \end{bmatrix}$$

**Rewards**:

The latent reward is as follows, with higher reward for more adherence.

$$R = \begin{bmatrix} 0 \\ 0.5 \\ 1 \end{bmatrix}$$

### F.2.3 HYPERPARAMETERS

Below we detail the hyperparameters used during the training.

| Hyperparameter | Value |
|---|---|
| K (Number of Epochs) | 300 |
| H (Horizon) | 1000 |
| $\epsilon$ (Epsilon) | $1 \times 10^{-5}$ |

Table 2: Hyperparameters used in Constructed Environment Training.

### F.2.4 EVOLUTION OF ESTIMATION ERRORS THROUGHOUT TRAINING

We monitor and visualize several key errors during the training process to ensure its integrity and effectiveness.

**index_error** This metric quantifies the disparity between the current index estimated by the (DOPL) algorithm and the optimal index derived from solving the optimization problem (6).

**F_error:** This denotes the discrepancy in preference estimation, measured as the root mean squared error between the estimated preference matrix $\hat{\mathbf{F}}^k$ at iteration $k$ and the ground truth preference matrix $\mathbf{F}$.

**P_error:** This error metric captures the deviation in transition kernel estimate $\hat{P}_n^k$ from the true kernel $P_n$.

**R_error:** This is the aggregate direct reward estimation error across all states and arms, i.e between $Q_n(s)$ and $\tilde{Q}_n(s)$, where $Q_n(s) := \ln \frac{\mathbf{F}_n^\star(\sigma_s, \sigma_{s\star})}{1 - \mathbf{F}_n^\star(\sigma_s, \sigma_{s\star})}$ and $\tilde{Q}_n(s) := \ln \frac{\tilde{\mathbf{F}}_n^\star(\sigma_s, \sigma_{s\star})}{1 - \hat{\mathbf{F}}_n^\star(\sigma_s, \sigma_{s\star})}$ where $s_\star$ is the reference state and $\star$ the reference arm and $\tilde{\mathbf{F}}$ is the estimated preference matrix with the upper confidence term added to it.

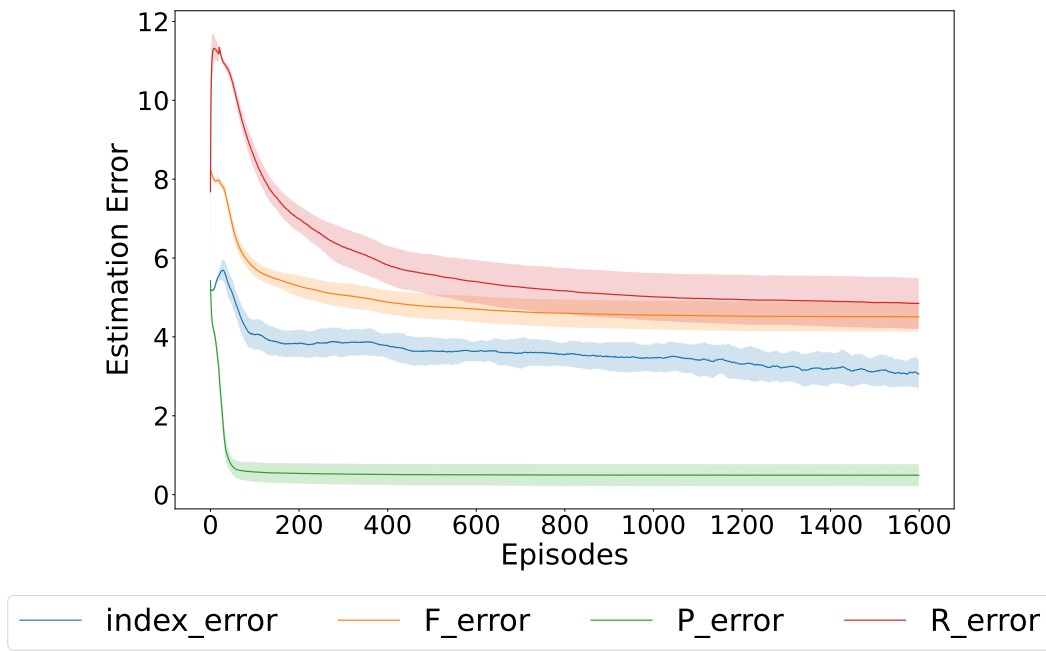

Figure 4: DOPL Estimation Errors During CPAP Training

## F.3 ARMMAN ENVIRONMENT

### F.3.1 OVERVIEW

For a detailed description refer to Appendix B.3. There are a total of 20 beneficiaries in this simulation each modeled by an MDP. These beneficiaries are of three types. Type A beneficiaries who are responsive to the treatment but need a little push. Type B beneficiaries are moderately responsive and Type C are less responsive to treatment. We ensure that a 1:1:3 ratio is maintained among the beneficiary types. The budget constraint allows intervention with a maximum of 10 beneficiaries at a given timestep.

### F.3.2 SYSTEM DYNAMICS

The dynamics of the system are such that each beneficiary type is associated with a transition matrix defined by specific probability ranges. For each beneficiary, we randomly sample transition probabilities within these ranges according to their type, ensuring that the resulting transition matrix is valid—that is, all probabilities are non-negative and each row sums to one.

**Type A beneficiaries    Transition Matrices:**

- **Action 0 (No intervention):**

$$P = \begin{bmatrix} 0.5 - 0.95 & 0 - 0.90 & 0.05 \\ 0.05 & 0 - 0.5 & 0.45 - 0.95 \\ 0.05 & 0.1 - 0.6 & 0.35 - 0.85 \end{bmatrix}$$

- **Action 1 (Intervention):**

$$P = \begin{bmatrix} 0.5 - 0.95 & 0 - 0.90 & 0.05 \\ 0.45 - 0.95 & 0 - 0.5 & 0.05 \\ 0.05 & 0.1 - 0.6 & 0.35 - 0.85 \end{bmatrix}$$

**Type B beneficiaries    Transition Matrices:**

- **Action 0 (No intervention):**

$$P = \begin{bmatrix} 0.5 - 0.95 & 0 - 0.90 & 0.05 \\ 0.05 & 0.1 - 0.6 & 0.35 - 0.85 \\ 0.05 & 0.1 - 0.6 & 0.35 - 0.85 \end{bmatrix}$$

- **Action 1 (Intervention):**

$$P = \begin{bmatrix} 0.5 - 0.95 & 0 - 0.90 & 0.05 \\ 0.15 - 0.65 & 0.3 - 0.8 & 0.05 \\ 0.05 & 0.1 - 0.6 & 0.35 - 0.85 \end{bmatrix}$$

**Type C beneficiaries    Transition Matrices:**

- **Action 0 (No intervention):**

$$P = \begin{bmatrix} 0.5 - 0.95 & 0 - 0.90 & 0.05 \\ 0.05 & 0.1 - 0.6 & 0.35 - 0.85 \\ 0.05 & 0.1 - 0.6 & 0.35 - 0.85 \end{bmatrix}$$

- **Action 1 (Intervention):**

$$P = \begin{bmatrix} 0.5 - 0.95 & 0 - 0.90 & 0.05 \\ 0.05 - 0.50 & 0.45 - 0.90 & 0.05 \\ 0.05 & 0.1 - 0.6 & 0.35 - 0.85 \end{bmatrix}$$

**Rewards**:

$$R = \begin{bmatrix} 1 \\ 0.5 \\ 0 \end{bmatrix}$$

### F.3.3   HYPERPARAMETERS

Below we detail the hyperparameters used during the training.

| Hyperparameter | Value |
|---|---|
| K (Number of Epochs) | 20000 |
| H (Horizon) | 5 |
| $\epsilon$ (Epsilon) | $1 \times 10^{-5}$ |

Table 3: Hyperparameters used in ARMMAN Environment Training.

