# OpenReview forum: "DOPL: Direct Online Preference Learning for Restless Bandits with Preference Feedback"
_ICLR.cc/2025/Conference — ICLR 2025 Poster_

### Official Review · Reviewer_mNbh · 2024-10-26

**Soundness:** 3
**Presentation:** 2
**Contribution:** 3
**Rating:** 6
**Confidence:** 4

**Summary:**

This paper studies the restless multi-armed bandit (RMAB) problem with preference feedback (PREF-RMAB) rather than direct reward estimation, motivated by real-world applications like app marketing and CPAP treatment. To address the problem that some arms in some states may not be visited frequently, the authors propose a new method to infer the empirical average of arm preference via the other arms’ empirical preference estimations. Additionally, the authors transform the original reward-based optimization problem into another one in terms of preference feedback directly. Using this transformation, the authors develop a low-complexity index policy for decision making. They also provide theoretical analysis of the regret bound, establishing a regret bound of $\tilde{\mathcal{O}}(\sqrt{T\ln T})$. Finally, the authors conduct experiments to verify the performance of their proposed DOPL algorithm.

**Strengths:**

1.	The authors present novel approaches to address the PREF-RMAB problem. The results in Lemmas 4 and 5 are particularly interesting and have the potential to inform future algorithmic design.
2.	The propose DOPL algorithm works well on the PREF-RMAB problem.
3.	I understand that analyzing the regret bound of the RMAB problem with preference feedback is challenging, so the inclusion of this theoretical bound is commendable.

**Weaknesses:**

1.	Although the authors make a strong effort to illustrate potential real-world applications of the PREF-RMAB problem, the justification remains unconvincing. For instance, in the app marketing scenario, users’ state transitions do not satisfy the memoryless Markov chain property, as a user currently in state $s_4$  cannot directly transition to $s_1$, and time tracking is ambiguous. Similar concerns apply to the other examples.
2.	The writing can be further improved. For example,

    (a) Adding more detail on the composition of the preference matrix $\mathbf{F}$ would improve clarity.

    (b) Eq. (2) needs to be improved, as the notation is confusing.

    (c) The objective function (6) is not easy to follow. Please define $\mu^{\pi}$ and $\mu_n$ first. I misunderstood it as a myopic problem until I saw the definition of mu^pi.

    (d) I think Lemmas 4 and 5 are more important than Lemmas 1 and 2. The authors can change them into propositions.

    (e) Lemma 2 can be improved by defining $Q_n(s)$ first and then present the result in Lemma 2.

**Questions:**

1.	The important results Lemmas 4 and 5 are based on Lemma 1. However, it is unclear why there is only a single reference arm * and a single reference state in Lemma 1. In the RMAB setting, DM selects B arms at each time slot, so the use of a single reference arm seems inconsistent.
2.	In Eq. (4), if $\epsilon=o(1)$, the confidence width becomes arbitrarily large. While if $\epsilon=\Theta(1)$, the probability becomes very small. How do the authors balance this trade-off? A more detailed discussion of the setting for $\epsilon$ would be helpful.

---

> ### Author Response · Authors · 2024-11-20
> **Official Response by Authors (1/3)**
>
> Thank you very much for your review and constructive comments, as well as giving the positive rating of our work. Here we would like to address the reviewer's concerns and hope that can help raise the rating of our paper.
>
> **Weakness \#1: ...Although the authors make a strong effort to illustrate potential real-world applications of the PREF-RMAB problem, the justification remains unconvincing...**
>
> **Response:** Thank you for your thoughtful comment. We appreciate the opportunity to clarify our intentions and the broader applicability of our  Pref-RMAB framework.
>
> Our proposed Pref-RMAB framework is designed to provide a general solution for any RMAB setting where only preference feedback is available, rather than scalar rewards. This distinguishes our work from conventional RMAB formulations that rely on well-defined reward signals (Whittle, 1988; Larrañaga et al.,2014; Bagheri \& Scaglione, 2015; Verloop, 2016; Zhang \& Frazier, 2021). By directly learning and making decisions based on pairwise preference data, our approach broadens the applicability of RMAB models to contexts where scalar reward estimation may be unavailable, or unreliable.
>
> The examples provided in the paper, such as app marketing and healthcare scenarios, serve to illustrate the potential applications and significance of our proposed setting where in practice, the preference feedback is more naturally rather than scalar rewards. Our primary goal is to demonstrate the applicability and versatility of Pref-RMAB in handling preference-based decision-making in such real-world applications. We acknowledge that in some real-world scenarios, including app marketing, user state transitions may appear to violate the memoryless Markov chain property due to complex dependencies and history effects. However, the Markovian assumption in our Pref-RMAB model serves as an abstraction that simplifies the modeling process while retaining key sequential decision-making dynamics.  More importantly, this assumption in practice can be reasonably approximated by defining states in such a way that captures sufficient historical context (e.g., aggregating past behaviors) to make state transitions approximately Markovian. For example, if a user in state
> $s_4$ cannot transition directly to $s_1$, the state space can be designed to capture intermediate states, transitions, or aggregate behaviors that reflect more realistic movement patterns while preserving Markovian dynamics for computational traceability.
>
> In practice, certain real-world applications may require additional modeling assumptions or extensions to approximate Markovian dynamics effectively. For instance, refining the state space or incorporating historical data may better capture complex transitions. While such adaptations can enhance the fidelity of specific applications, addressing these complexities falls outside the primary scope of this paper. Our focus is on establishing and analyzing the theoretical foundation and performance of the Pref-RMAB framework under the core assumption of Markovian dynamics for tractability.
>
> **Weakness \#2(a): Adding more detail on the composition of the preference matrix would improve clarity.**
>
> **Response:** We agree with the reviewer and the definition the preference matrix is important for the readers to understand preference learning in Pref-RMAB. Due to space constraints and to help readers better understand this concept, we indeed provided a detailed definition of the preference matrix along with a toy example in Appendix C (lines 942-980).
>
>
> **Weakness \#2(b): Eq. (2) needs to be improved, as the notation is confusing.**
>
> **Response:** Equation (2) is the definition of the preference feedback $\alpha(s_m^t,s_n^t)$ between arm $m$ in state $s_m^t$ and arm $n$ in state $s_n^t$. Note that this Bernoulli random variable $\alpha(s_m^t,s_n^t)$ is drawn according to the widely-used Bradley-Terry (BT) model with the detail given in Equation (2). In the popular dueling bandits, each arm is "stateless" and hence the expression corresponding to the BT model is relatively simpler. However, in our Pref-RMAB, each restless arm is stateful which somehow "complicates" the preference matrix as indicated in the preference matrix (see response above), and hence further complicates the definition of Equation (2) correspondingly. Nevertheless, as defined in lines 126-128, "Let $\sigma$ be a permutation on $\mathcal{A},$ and $\sigma_s$ be the position of element $s$ in $\mathcal{A}.$" Thus, the comparison between arm $m$ in state $s_m^t$ and arm $n$ in state $s_n^t$ corresponds to the row $(m-1)|\mathcal{S}|+\sigma_{s_{m}^t}$ and the column $(n-1)|\mathcal{S}|+\sigma_{s_{n}^t}$ in the preference matrix $\mathbf{F}.$ This leads to the first equality in Equation (2), and the second equality is directly from the definition of the BT model. We hope this clarifies the confusion, and if you have other concerns, we are happy to engage further.

---

> ### Author Response · Authors · 2024-11-20
> **Official Response by Authors (2/3)**
>
> **Weakness \#2(c): The objective function (6) is not easy to follow. Please define $\mu^\pi$ and $\mu_n$ first. I misunderstood it as a myopic problem until I saw the definition of $\mu^\pi$.**
>
> **Response:** Thank you for this suggestion. We have modified the paper to define $\mu_n$ and then $\mu^\pi$ first. We highlight the changes in blue.
>
> **Weakness \#2(d): I think Lemmas 4 and 5 are more important than Lemmas 1 and 2. The authors can change them into propositions.**
>
> **Response:** Thank you for your suggestions. We have changed them into Proposition 1 and Proposition 2. We highlight the changes in blue.
>
> **Weakness \#2(e): Lemma 2 can be improved by defining first and then present the result in Lemma 2.**
>
> **Response:** Thank you for your suggestion. We modified the paper by defining the preference-reference term before presenting Lemma 2 (now Proposition 2). We highlight the changes in blue.
>
> **Question \#1: The important results of Lemmas 4 and 5 are based on Lemma 1. However, it is unclear why there is only a single reference arm $*$ and a single reference state in Lemma 1. In the RMAB setting, DM selects B arms at each time slot, so the use of a single reference arm seems inconsistent.**
>
> **Response:** Thank you for your thoughtful comment regarding the use of a single reference arm and state in Lemma 1. However, we are afraid that there is a misunderstanding here regarding the reference arm/state and the design of our DOPL algorithm. We appreciate the opportunity to clarify this design choice and its significance in addressing the core challenges of the Pref-RMAB framework.
>
> A key challenge in the \textsc{Pref-RMAB} setting is that the decision-maker (DM) does not have access to scalar rewards for each arm and state; instead, only pairwise preference feedback is available. This makes direct optimization using conventional RMAB methods difficult, as they typically rely on scalar reward values for decision-making and policy evaluation, as the LP in Eqs. (6)-(9) in Section 3.3. Lemma 1 plays a critical role in bridging this gap by establishing a strong connection between scalar rewards and preference feedback. Specifically, it shows that the scalar reward for any arm $n$ in a state $s$ can be represented in terms of the preference feedback with respect to a fixed reference arm $\star$ and a reference state $s_\star$. As stated in Lemma 2 and Remark 3, we therefore can define the ``preference-reference" $Q_n(s)$ to fully represent the preference of arm $n$ in state $n$ in our Pref-RMAB framework.
>
>
> You point out that the DM needs to select $B$ arms at each time slot. We formulate this problem in Section 3.3 as defined in Equations (6) - (9) where the feedback of each arm $n$ in state $s$ is the scalar reward $r_n(s)$. One key technical contribution in this paper is Lemma 5, where we showed that we can fully transform the objective (6) into an objective in terms of preference feedback. More importantly, the reference arm and state can be any arm and state, as long as they remain fixed throughout the estimation process. This flexibility does not constrain the DM's ability to select multiple arms at each decision epoch but rather serves as a necessary construct to transform preference feedback into a meaningful scalar reward representation.
>
> Given this fixed reference arm $\star$ and reference state $s_\star$, DOPL only learns one specific column of the preference matrix $\mathbf{F}$, i.e., $\mathbf{F}(: ,(\star-1)|\mathcal{S}|+\sigma_{s_\star})$, corresponding to the preference between the reference state $s_\star$ of reference arm $\star$ with any arm $n\in\mathcal{N}$ in any state $s\in\mathcal{S}$,  as discussed in Section 3.2 (lines 272-279). This substantially reduces the computation complexity as well. Although the DM selects $B$ arms at each time slot and the reference arm $ \star$ is not selected, we can still infer the values in $\mathbf{F}(: ,(\star-1)|\mathcal{S}|+\sigma_{s_\star})$ according to another novel theoretical result in Lemma 4. In summary, the fixed $\star$ and $s_\star$ serve as a bridge to transform the scalar-reward based optimization problem in Equations (6)-(9) to the preference-feedback-based optimization in Eq. (10).

---

> > ### Author Response · Authors · 2024-11-20
> > **Official Response by Authors (3/3)**
> >
> > **Question \#2: In Eq. (4), if $\epsilon=o(1)$, the confidence width becomes arbitrarily large. While if  $\epsilon=\Theta(1)$
> > , the probability becomes very small. How do the authors balance this trade-off? A more detailed discussion of the setting for $\epsilon$ would be helpful.**
> >
> > **Response:** Thank you for raising this important point about the parameter $\epsilon$ in Eq. (4) and its impact on the confidence width. You are correct to note that there is a trade-off: if $\epsilon$ is small, the confidence width becomes large, and if $\epsilon$ is large, the probability of containing the true parameter becomes small. However, it is known to have a rigorous proof that as long as $\epsilon\in (0,1)$, the theorem always holds, ensuring the validity of our theoretical guarantees.
> >
> > This type of trade-off framework is widely used within the UCRL (Upper Confidence Reinforcement Learning) framework (Akbarzadeh et al. 2022; Jaksch et al. 2010; Wang et al. 2020), where the balance between exploration and exploitation is controlled by the choice of $\epsilon$. The trade-off operates as follows: If $\epsilon$ is smaller, the upper bound of the confidence width is larger, but the probability of the true parameter lies within this interval is higher, resulting in more conservative exploration. If $\epsilon$ is larger, the upper bound is smaller, which narrows the confidence interval, but this comes with a lower probability of covering the true parameter, leading to more aggressive exploitation.
> >
> > To achieve a balance between these extremes, we select $\epsilon$ as a function of the time horizon
> > $T:=KH$, ensuring that it scales appropriately with the number of episodes
> > $K$
> > and decision epochs
> > $H$. A common and effective approach is to set
> > $\epsilon$ to decay slowly over time, such as
> > $\epsilon=\Theta(1/T)$. This ensures that the confidence width gradually becomes narrower as more data is collected, reflecting increasing certainty about the transition dynamics without overly restricting exploration.
> > In particular, Akbarzadeh et al. 2022 uses
> > $\epsilon=1/T$, achieving a balance that maximizes learning efficiency.
> > Jaksch et al. 2010 adopt a slightly different scaling, with
> > $\epsilon=1/3T$, which reflects a slightly more conservative trade-off in exploration-exploitation dynamics.
> >
> > The choice of $\epsilon$ directly affects the theoretical regret bounds of our algorithm. By ensuring that
> > $\epsilon$ decays appropriately, we can maintain a balance between maintaining sufficient exploration (to guarantee sublinear regret) and minimizing unnecessary exploration (to improve convergence rates). Our theoretical analysis guarantees that the regret remains sublinear by properly tuning
> > $\epsilon$ within this trade-off space.

---

> > > ### Comment · Reviewer_mNbh · 2024-11-20
> > >
> > > I thank the authors' response. I understand that this work focuses on the theoretical analysis and I acknowledge the theoretical contributions of this work. However, I remain unconfident about the scope of real-world applications for Pref-RMAB. As such, I will maintain my score, weakly supporting the authors.

---

> > > > ### Author Response · Authors · 2024-11-25
> > > > **Further clarification**
> > > >
> > > > We thank the reviewer again for acknowledging our theoretical contributions. We would like to further clarify the rationality of our Pref-RMAB framework and why it is a "better or more realistic" model than the "standard" RMAB for many real-world applications.
> > > >
> > > > Since Whittle (1988) proposed the "standard" RMAB framework (Section 2.1), it has been extensively used to model many real-world applications with constrained sequential decision-making problems, from job scheduling (Bertsimas et al. 2000, Yu et al. 2018), cloud computing (Borkar et al. 2017, Xiong et al. 2023), online advertisement (Meshram et al. 2016) to healthcare (Killian et al. 2021, Mate et al. 2021). Despite the wide-range application, **a key limitation or the success** of the standard RMAB is that it **implicitly assumes** that the decision-maker in these real-world applications can always receive **an exact/perfect scalar reward feedback**. Unfortunately, specifying an exact reward function in practice can be very challenging, which may vary over different applications or even change over different scenarios for the same application. Exacerbating this challenge is the fact that obtaining the exact scalar reward feedback may be even infeasible in practice, which is especially pronounced in online advertisement/recommendation systems and healthcare applications. To bridge such a gap between the intrinsic nature of many real-world applications (only preference feedback is available) and the standard RMAB model (requiring scalar reward feedback), we advocate a new model named Pref-RMAB, in which the decision-maker makes online decisions purely relying on the preference feedback. **Therefore, for any existing/studied RMAB problem or real-world applications (aforementioned ones and references in lines 37-44) that can be modeled as a RMAB, as long as the preference feedback is more natural or much easier to be accessed than the scalar reward, they can be modeled by our Pref-RMAB framework and solved by our DOPL algorithm.**
> > > >
> > > > Of course, as we responded to the Weakness \#1, there often exists a bit discrepancy between the theoretical modelings and the real-world scenario, and our proposed DOPL algorithm also has some limitations (discussed in Section 6), **this work takes the first step to address the long-existing limitations of the standard RMAB** and tackles many new technical challenges in the algorithm design and the performance analysis in Pref-RMAB, which are the key contributions of this work as acknowledged by the reviewer. Like the more recent popular RLHF framework (advancing the standard RL framework), we believe that Pref-RMAB will benefit the community to study online sequential decision-making problems under instantaneous constraints for many emerging real-world applications.

---

> > > > > ### Comment · Reviewer_mNbh · 2024-11-26
> > > > >
> > > > > I respectfully disagree with the authors' claim that Pref-RMAB is superior to or more realistic than the standard RMAB. To some extent, Pref-RMAB can be viewed as a variant of the standard RMAB, as many applications do not require human feedback. For instance, in transportation networks, path conditions can transition between states according to a Markov chain, and the travel cost of each path can be characterized by its latency. Similar scenarios arise in applications like cognitive radio networks. Therefore, I remain unconvinced about the broader applicability of Pref-RMAB. While RLHF is popular due to its relevance to LLM applications involving direct human interaction, practical issues such as reward hacking still persist in RLHF.

---

> > > > > > ### Author Response · Authors · 2024-11-27
> > > > > > **Thank you for the follow-up comment**
> > > > > >
> > > > > > We agree with the reviewer that the standard RMAB has been extensively used to model many real-world applications including the examples pointed out by the reviewers, where the scalar reward feedback is feasible. On the other hand, there are also many emerging applications such as online advertisement, recommendation systems and healthcare, where preference feedback is more natural and it is often hard to specify an exact reward function.  Thus, comparing to the standard RMAB, our Pref-RMAB is a more appropriate model for such applications. That's why we claim that **"for any existing/studied RMAB problem or real-world applications that can be modeled as a RMAB, as long as the preference feedback is more natural or much easier to be accessed than the scalar reward, they can be modeled by our Pref-RMAB framework and solved by our DOPL algorithm."**

---

### Official Review · Reviewer_FhNw · 2024-11-04

**Soundness:** 3
**Presentation:** 2
**Contribution:** 3
**Rating:** 6
**Confidence:** 4

**Summary:**

This paper proposes the PREF-RMAB model, which observes only the preference between two arms rather than the exact reward of each arm in the restless multi-armed bandit problem. By expressing the reward function of any arm n  in any state as the sum of a reference reward and a function related to the preference probability between arm n in state s and a reference arm in a reference state, the authors develop a direct index policy based on preference data to choose which arms to activate in the online manner. They establish an O (√(NT|S|ln⁡〖(|S||A|NT)〗 )) regret bound for the proposed algorithm.

**Strengths:**

1. Although RLHF has recently gained significant attention due to its applications in large language models and robotics, this work is the first to consider a preference-based reward model in the restless bandit problem, opening the door for RLHF to be applied much more broadly.

2. By establishing a connection between pairwise preferences and reward values, the authors transform the reward value of each arm and state into a measure based on the preference probability between this state and a reference arm and state, which is intriguing. Additionally, the algorithm can infer the preference between the element j in the preference matrix \(\mathbf{F}\) and the reference elements \(s_*\) without directly comparing them, but rather through an intermediate comparator. This clever design reduces the complexity to that of directly observing the reward.

**Weaknesses:**

•	1. A question remains as to whether a preference-based model can outperform direct reward estimation, and whether we really need the preference-base model in RMAB problem. In the first two examples presented in the paper, APP MARKETING and CPAP TREATMENT, while reward data is challenging to estimate accurately and may contain substantial noise, it can still be estimated in some form. Conversely, since preference data inherently provides less information, it is unclear whether incorporating preference data can improve performance over direct reward estimation. Most papers on RLHF use preference feedback to shape the reward for different trajectories in robotics or large language models (LLMs), where trajectory rewards are inherently complex and require function approximation methods to estimate the reward function. However, the RMAB model studied in this paper is a tabular MDP, where rewards can be estimated through multiple sampling.

•	2. In Algorithm 3 of Section C.3, at each step \( h \), the algorithm performs \( (B-1) \) random duels to obtain the preference data. Then, when constructing the preference between \(s_n\) and the reference \(s_*\), only the preference between \(s_n\) and \( j \) and \(s_*\) is used. It appears that \( s_n \) could be compared with many other columns \( i \) in the \( F \) matrix, but the algorithm does not leverage this data. Consequently, Algorithm 3 makes inefficient use of the available information.

•	3. The MDP considered in the paper is essentially a tabular MDP, and the regret scales with the square root of \( |S| \) (the size of the state space), which may be inefficient for large state spaces.

**Questions:**

•	1. In the 11th step of Algorithm 3 in Section C.3, when inferring \(\hat{\mathbf{F}}_n^{*,k+1,\text{inf}}(\sigma_{s_n}, \sigma_{s_*})\), only one intermediate column \( j \) in \(\mathbf{F}\) is selected to compute \(\hat{\mathbf{F}}_n^{*,k+1,\text{inf}}(\sigma_{s_n}, \sigma_{s_*})\) based on \(\hat{\mathbf{F}}_n^{*,k}(\sigma_{s_n}, j)\) and \(\hat{\mathbf{F}}_n^{*,k}(j, ((*-1)|S|+\sigma_{s_*}))\). However, after selecting \( B \) arms at each step, the duels are conducted randomly, resulting in many comparison data points beyond \( j \) that are not used in the inference process. Could this lead to substantial unutilized information? Could more comparison data, beyond just \( j \), be leveraged to infer the preference between \( s_n \) and \( s_* \)? Alternatively, rather than performing duels randomly, might strategically choosing duel pairs improve performance?

•	2. In Eq. (3), the authors define \(\pi^{opt}\) as the solution of (1) with scalar rewards, but the footnote states that the regret is with respect to preference feedback, which seems contradictory. This part is unclear to me.

•	3. In Eq. (46), the suboptimality is accumulated over \( K \) episodes. However, since \( \omega^k_n \) is a probability measure and \( Q_n(s) \) represents the relative reward of arm \( n \) in state \( s \), which involves the reward incurred at a specific step \( h \) within episode \( k \), why doesn’t \( h \) appear in this decomposition?

•	4. In Lemma 6, the authors use Lemma 11 to argue that term0 is negative. However, I find this reasoning unclear, as \({\pi^*}\) does not appear to align with \(\tilde{\pi}\) as defined in Lemma 6. Specifically, \(\mu_{\pi^*}\) represents the optimal solution of Eq. (6)-(9), while \(\tilde{\pi}\) is the index policy developed from \(\mu_{\pi^*}\) to satisfy the hard constraint. Therefore, I am uncertain that Lemma 11 can indeed be used to prove Lemma 6, and concluding that term0 is negative is not straightforward.

•	5. In the proof procedure following Eq. (49), from the fourth to the fifth line, the inequality \(\sum_{k=1}^K\sqrt{\frac{1}{Z^k_n(s,a)}} \leq \sqrt{Z^K_n(s,a)}\) appears incorrect. For instance, if the algorithm visits arm \( a \) at state \( s \) once at the beginning and never revisits this state, it would hold that \( Z^1_n(s,a) = \dots = Z^K_n(s,a) = 1 \), yielding \(\sum_{k=1}^K\sqrt{\frac{1}{Z^k_n(s,a)}} = K\), which is indeed greater than \( \sqrt{Z^K_n(s,a)} = 1\). If I have misunderstood this part, please clarify.

•	6. In the sentence between lines 1398 and 1399, I think the statement \(\sum_{n=1}^N\sum_{(s,a)}Z^T_n(s,a) \leq NT\) should instead be \(\sum_{n=1}^N\sum_{(s,a)}Z^T_n(s,a) = T\), as the total visits across all arms and states should sum to \(T\). In fact, only under this revised statement can the inequality (c) above this sentence be satisfied, otherwise it should be \(\sum_{n=1}^N\sqrt{\sum_{(s,a)}Z_n^K(s,a)}\leq N\sqrt{T}\). This confusion also appears in the sentence from 1503 to 1505. If I have misunderstood this part, please clarify.

---

> ### Author Response · Authors · 2024-11-20
> **Official Response by Authors (1/5)**
>
> Thank you very much for your review and constructive comments. Here we would like to address the reviewer's concerns and hope that can help raise the rating of our paper.
>
> **Weakness \#1:  A question remains as to whether a preference-based model can outperform direct reward estimation, and whether we really need the preference-base model in RMAB problem...**
>
> **Response:** Thank you for your comment. We would like to explain why reward estimation performs worse than preference feedback numerically and qualitatively.
>
> In our numerical results, as presented in Figures 1, 2, and 3, we compare the DOPL with three benchmark algorithms
> MLE_WIBQL, MLE_QWIC and MLE_LP, which rely on maximum likelihood estimation (MLE) to convert preference feedback into scalar reward estimates before applying standard RL algorithms (e.g., Whittle-index-based Q-learning for MLE_WIBQL,  MLE_QWIC and linear programming for MLE_LP). As shown in Figures 1, 2, and 3 (left), DOPL significantly outperforms all considered
> baselines and reaches close to the oracle. In addition, DOPL is much more computationally efficient
> since it can directly leverage the preference feedback for decision making, while the considered
> baselines require a reward estimation step via MLE in the presence of preference feedback, which
> can be computationally expensive. Finally, our DOPL yields a sublinear regret as illustrated in
> Figures 1, 2, 3 (right), consistent with our theoretical analysis (Theorem 1), while none of these
> baselines yield a sublinear regret in the presence of preference feedback.
>
> The underlying reason is that
> these indirect approaches inherently introduce noise and estimation errors during the transformation from preference-based data to scalar rewards.
> Converting preference feedback into scalar rewards typically assumes that an accurate mapping function exists. However, in complex dynamic environments modeled by the \textsc{Pref-RMAB} framework, such mappings can be non-linear, unknown, or resource-intensive to approximate correctly. As those benchmark algorithms need to design index-based policies (such as Whittle index, and LP-based index), which can be very sensitive to the estimation error. Thus, the cumulative effect of these inaccuracies during each decision epoch leads to suboptimal action selection, resulting in higher regret. In contrast, DOPL circumvents this issue by directly learning from preference feedback, eliminating the need for a reward transformation step. This direct approach allows DOPL to leverage preference data more effectively for decision-making, reducing the impact of noise and achieving sublinear regret.
>
> Furthermore, as we discussed in lines 69-82, "a straightforward method as inspired by RLHF is to learn a scalar reward function to represent the preference feedback of humans, and then apply existing RL algorithms for RMAB with this estimated reward function to the \textsc{Pref-RMAB} problem." However, the downside of directly applying this RLHF method to \textsc{Pref-RMAB} is its complexity and insufficiency. "In view of these issues, there is an innovative line of work that directly learns from preferences without explicit reward modeling, such as the popular direct preference optimization (DPO)".  However, most of these RLHF or DPO methods are **offline**, leading to the issues of overoptimization, while the decision maker in our \textsc{Pref-RMAB} interacts with the environment in an **online** manner. We kindly refer the reviewer to lines 69-82 for more detailed discussions.
>
> Last but not least, the context of LLMs significantly differs from our RMAB settings. In RMAB, each "restless" arm is a MDP with state transitions and rewards being well-defined (see Section 2.1). In contrast, LLM outputs and the associated feedback are highly context-dependent and subjective. Therefore, despite that both our \textsc{Pref-RMAB} and LLMs consider "preference feedback", the settings and intrinsic nature differ dramatically, and hence are not quite comparable. To our best knowledge, LLMs cannot be formulated as a RMAB problem, and hence our RMAB cannot be viewed as a LLM.

---

> > ### Author Response · Authors · 2024-11-20
> > **Official Response by Authors (2/5)**
> >
> > **Weakness \#2:  In Algorithm 3 of Section C.3, at each step $h$, the algorithm performs $B-1$ random duels to obtain the preference data. Then, when constructing the preference between $s_n$ and the reference $s_*$, only the preference between $s_n$ and $j$ and $s_*$ is used. It appears that $s_n$ could be compared with many other columns $i$ in the $F$ matrix, but the algorithm does not leverage this data. Consequently, Algorithm 3 makes inefficient use of the available information.**
> >
> > **Response:** We are afraid that there is a misunderstanding here about the design merit of our DOPL. In contrast to the reviewer's comment, this is actually one key part we intentionally designed in our proposed DOPL algorithm to make it "efficient". We would like to elaborate it in details below.
> >
> > Indeed, it is theoretically possible to leverage all sampled pairs for comparison in each time slot (decision epoch). This will result in a total of
> > $B(B-1)/2$ comparisons in each time slot. Since it leverages all possible comparisons between the $B$ activated restless arms in each time slot, this would greatly maximize the information obtained from each interaction, potentially providing a more comprehensive view of the preference matrix
> > $F$. However, in many real-world applications, such as healthcare and online marketing, **performing such a large number of duels or comparisons in each time slot is highly cost-prohibitive, impractical or even infeasible.** Comparisons often require human effort, expertise, and time, making them expensive and difficult to scale. In such settings, reducing the number of comparisons without sacrificing performance is critical for the algorithm's practicality and usability.
> > One primary design principle in our DOPL is to decrease the number of comparisons required in each time slot while still achieve robust performance.
> >
> > As we discussed in Remark 4, we only need to learn half (e.g. upper triangular part) of the preference matrix $F$. More importantly,  although we leveraged the BT model, its success in Pref-RMAB would require the decision maker (DM) to activate any arm in any state frequently (see lines 206-215), which often, in practice, are hardly feasible. A key contribution in our design is the “preference inference” (Step 3 in Section 3.2). Thanks to this design (its error is guaranteed by Lemmas 1,2,4), the actual implementation of our DOPL only requires $B-1$
> >  comparison at each time slot, rather than $B(B-1)/2$ comparisons. In addition, in practice,
> > $B\ll N$, this level of preference feedback is practical and obtainable in many real-world applications such as healthcare and resource allocation as mentioned earlier. This reduction in the number of comparisons is central to making DOPL more feasible for real-world applications.
> >
> > We give a toy example to illustrate why our DOPL only needs $B-1$ comparisons rather than $B(B-1)/2$ comparisons in each time slot.  Suppose that there are $N=10$ arms and the DM can pull $B=4$ restless arms at each time slot. For instance, arms 1, 2, 3, 4 are pulled, and we can request pairwise comparisons between arms (1, 2); (1, 3); (1, 4); (2,3); (2,4) and (3,4). As such, there are total 6 comparisons, i.e., $B(B-1)/2$
> > . However, the actual implementation of our DOPL only needs comparisons between arms (1, 2); (2,3) and (3,4), i.e., only 3 (i.e., $B-1$
> > ) comparisons. This is because the DM can infer the other comparisons, i.e., (1, 3), (1, 4), (2, 4),  based on our proposed “preference-inference” step.

---

> > > ### Author Response · Authors · 2024-11-20
> > > **Official Response by Authors (3/5)**
> > >
> > > **Weakness \#3: The MDP considered in the paper is essentially a tabular MDP, and the regret scales with the square root of $|S|$ (the size of the state space), which may be inefficient for large state spaces.**
> > >
> > > **Response:** You are right that we focus on a tabular setting in this paper, and the same tabular MDP setting has been extensively considered in existing reinforcement learning (RL) literature. Our goal is to propose a new Pref-RMAB framework and design a novel DOPL algorithm, and then establish strong theoretical foundations and rigorously demonstrate the validity and performance guarantees of our proposed DOPL algorithm. The tabular setting allows for precise analysis of regret bounds and a detailed characterization of learning behavior in environments with finite state and action spaces.
> > >
> > > Additionally, we acknowledge that achieving regret that scales with the square root of
> > > $|\mathcal{S}|$ is not an unfavorable result. In fact, this is among the best possible orders achievable in this context, as demonstrated by existing works such as Akbarzadeh et al. (2022), Jaksch et al. (2010), and Wang et al. (2020), and many other RL literature.  We can also observe this from papers as [1-3]. These results highlight the inherent difficulty of RL in tabular MDPs and underscore the competitiveness of our approach relative to state-of-the-art methods.
> > >
> > >
> > > We also recognize that extending our approach to non-tabular settings, where state spaces can be large or continuous, would require incorporating function approximation techniques, such as linear function approximation or neural networks approximation. This is a promising direction for future work, as these methods can generalize across states and mitigate the dependence on the size of the state space, making our algorithm scalable to more complex and high-dimensional problems.
> > >
> > > [1] Jin, Chi, et al. "Provably efficient reinforcement learning with linear function approximation." Conference on learning theory. PMLR, 2020.
> > >
> > > [2] Azar, Mohammad Gheshlaghi, Ian Osband, and Rémi Munos. "Minimax regret bounds for reinforcement learning." In International conference on machine learning, pp. 263-272. PMLR, 2017.
> > >
> > > [3] Kwon, J., Efroni, Y., Caramanis, C. and Mannor, S., 2021. Rl for latent mdps: Regret guarantees and a lower bound. Advances in Neural Information Processing Systems, 34, pp.24523-24534.

---

> > > > ### Author Response · Authors · 2024-11-20
> > > > **Official Response by Authors (4/5)**
> > > >
> > > > **Question \#1: In the 11th step of Algorithm 3 in Section C.3, when inferring...choosing duel pairs improve performance?**
> > > >
> > > > **Response:** Once again, we are afraid that there is a misunderstanding on the design of our DOPL algorithm. We kindly refer the reviewer to our above response to the **Weakness \#2**.
> > > >
> > > > **Question \#2:  In Eq. (3), the authors define $(\pi^{opt})$ as the solution of (1) with scalar rewards, but the footnote states that the regret is with respect to preference feedback, which seems contradictory. This part is unclear to me.**
> > > >
> > > >
> > > > **Response:** In lines 150-151, we stated that ``For ease of presentation, we refer to (1) as the optimization problem that the DM needs to solve in PREF-RMAB in the rest of this paper." $\pi^{opt}$ is the optimal index policy when the true preference feedback is available based on the true reward signal, while $\\{\pi^k\\}_{k=1}^K$ is designed based on learned preference feedback. Hence the regret is defined with respect to preference feedback.
> > > >
> > > > **Question \#3: In Eq. (46), the suboptimality is accumulated over $( K )$ episodes. However, since $( \omega^k_n )$ is a probability measure and $( Q_n(s) )$ represents the relative reward of arm $( n )$ in state $( s )$, which involves the reward incurred at a specific step $( h )$ within episode $( k )$, why doesn’t $( h )$ appear in this decomposition?**
> > > >
> > > > **Response:**  Notice that $\omega^k_n $ is a probability measure and $Q_n(s)$ represents the preference-reference of arm $n$ in state $s$, which denotes the expected reward  within the entire episode $k$. It is not defined for a single time step $h$, and thus $h$ is not included in Eq. (46).
> > > >
> > > > **Question \#4:  In Lemma 6, the authors use Lemma 11 to argue that term0 is negative...**
> > > >
> > > > **Response:** Once again, we are afraid that there is a misunderstanding here. We would like to reiterate the definitions of some key terms. Notice that $\\{\tilde{\pi}^k, \forall k\\}$ is the direct index policy executed in each episode $k$ with transition functions drawn from the confidence interval, and
> > > > $J(\\{\tilde{\pi}^k, \forall k\\}, \\{\tilde{\mathbf{F}}^k, \forall  k\\}, T)$ denotes the total rewards achieved by policy $\\{\tilde{\pi}^k, \forall k\\}$ with overestimated preference $\\{\tilde{\mathbf{F}}^k, \forall k\\}$. This represents the fact that we are constructing a **virtual expanded system** that contains the original true system, and employ a virtual optimism policy on that virtual expanded system. Since the virtual expanded system contains the original system, the performance of the $\\{\tilde{\pi}^k, \forall k\\}$ in the virtual optimism system is no worse than that of the optimal direct index policy to the original system. This is called **Optimism**, and has been leveraged in many existing work (Wang et al., 2020; Xiong et al., 2022b) and UCRL (Upper Confidence Reinforcement Learning) framework (Akbarzadeh et al, 2022; Jaksch et al, 2010). As to why the true system is included in the optimism system, it is due to the Hoeffding inequality according to Lemma 3 and Lemma 9, i.e., the true transition kernels lies inside of the confidence ball with probability $1-2\epsilon.$

---

> > > > > ### Author Response · Authors · 2024-11-20
> > > > > **Official Response by Authors (5/5)**
> > > > >
> > > > > **Question \#5: In the proof procedure following Eq. (49), from the fourth to the fifth line, the inequality $(\sum_{k=1}^K\sqrt{\frac{1}{Z^k_n(s,a)}} \leq \sqrt{Z^K_n(s,a)})$ appears incorrect. For instance, if the algorithm visits arm $( a )$ at state $( s )$ once at the beginning and never revisits this state, it would hold that $( Z^1_n(s,a) = \dots = Z^K_n(s,a) = 1 )$, yielding $(\sum_{k=1}^K\sqrt{\frac{1}{Z^k_n(s,a)}} = K)$, which is indeed greater than $( \sqrt{Z^K_n(s,a)} = 1)$. If I have misunderstood this part, please clarify.**
> > > > >
> > > > > **Response:**
> > > > > Thank you for your comments. This is not a very challenging derivation and hence some details were not fully incorporated in the paper. Here we would like to clearly explain how we can derive these inequalities.
> > > > >
> > > > > First, we would like to introduce a simple inequality, which is a known result.  For any sequence of numbers $w_1,w_2,...,w_T$ with $0\leq w_k$, define $W_{k}:=\sum_{i=1}^k w_i$, then we have
> > > > >     \begin{align*}
> > > > >         \sum_{k=1}^T \frac{w_k}{\sqrt{W_{k}}} \leq (1+\sqrt{2}) \sqrt{W_T}.
> > > > >     \end{align*}
> > > > > We can briefly show why it works.
> > > > > The proof follows by induction.
> > > > >
> > > > > When $t=1$, it is true as $1\leq \sqrt{2}+1$.
> > > > > Assume for all $k\leq t-1$, the inequality holds, then we have the following:
> > > > > $$
> > > > >  \sum_{k=1}^T \frac{w_k}{\sqrt{W_{k}}}= \sum_{k=1}^{T-1} \frac{w_k}{\sqrt{W_{k}}} + \frac{w_T}{\sqrt{W_{T}}}
> > > > > $$
> > > > > $$
> > > > > \leq (1+\sqrt{2}) \sqrt{W_{T-1}} + \frac{w_T}{\sqrt{W_{T}}}
> > > > > $$
> > > > > $$
> > > > > =\sqrt{{(1+\sqrt{2})}^2 W_{T-1} + 2(1+\sqrt{2})w_T \sqrt{\frac{W_{T-1}}{W_T}} + \frac{{w_T}^2}{W_T}}
> > > > > $$
> > > > > $$
> > > > > \leq \sqrt{{(1+\sqrt{2})}^2 W_{T-1} + 2(1+\sqrt{2})w_T \sqrt{\frac{W_{T-1}}{W_{T-1}}} + \frac{{w_T}W_T}{W_T}}
> > > > > $$
> > > > > $$
> > > > > = \sqrt{{(1+\sqrt{2})}^2 W_{T-1} + (2(1+\sqrt{2})+1)w_T}
> > > > > $$
> > > > > $$
> > > > >  = (\sqrt{2}+1)\sqrt{(W_{T-1}+ w_T)}
> > > > > $$
> > > > > $$
> > > > > = (\sqrt{2}+1)\sqrt{W_T}.
> > > > > $$
> > > > > Second, based on the above inequality, we have
> > > > > $$
> > > > > \sum_{k=1}^{K}\sum_{n=1}^N\sum_{s,a} \sqrt{\frac{1}{2Z^k_n(s,a)}\ln\frac{4|\mathcal{S}||\mathcal{A}|NT}{\epsilon}}
> > > > > $$
> > > > > $$
> > > > > \leq \sqrt{\ln\frac{4|\mathcal{S}||\mathcal{A}|NT}{\epsilon}}\sum_{t=1}^{T}\sum_{n=1}^N\sum_{(s,a)}\mathbf{1}(s_n(t)=s,a_n(t)= a) \sqrt{\frac{1}{2Z_n^{t}(s,a)}}
> > > > > $$
> > > > > $$
> > > > > = \sqrt{\ln\frac{4|\mathcal{S}||\mathcal{A}|NT}{\epsilon}}\sum_{t=1}^{T}\sum_{n=1}^N\sum_{(s,a)}\frac{\mathbf{1}(s_n(t)=s,a_n(t)= a)}{ \sqrt{{2Z_n^{t}(s,a)}}}
> > > > > $$
> > > > > $$
> > > > > \leq  2\sqrt{\ln\frac{4|\mathcal{S}||\mathcal{A}|NT}{\epsilon}}\sum_{n=1}^N\sum_{(s,a)}{ \sqrt{{Z_n^{T}(s,a)}}}~~~\text{the inequality introduced above}
> > > > > $$
> > > > > $$
> > > > > \leq  2\sqrt{\ln\frac{4|\mathcal{S}||\mathcal{A}|NT}{\epsilon}}\sum_{n=1}^N{ |\mathcal{S}||\mathcal{A}|\sqrt{\frac{\sum_{(s,a)}{Z_n^{T}(s,a)}}{|\mathcal{S}||\mathcal{A}|}}}~~~\text{Jensen's inequality}
> > > > > $$
> > > > > $$
> > > > > \leq  2\sqrt{\ln\frac{4|\mathcal{S}||\mathcal{A}|NT}{\epsilon}}\sum_{n=1}^N{ \sqrt{{|\mathcal{S}||\mathcal{A}|}T}}~~~\text{due to} \sum_{s,a}Z_n^T(s,a)\leq T
> > > > > $$
> > > > > $$
> > > > > \leq  2N\sqrt{\ln\frac{4|\mathcal{S}||\mathcal{A}|NT}{\epsilon}}{ \sqrt{{|\mathcal{S}||\mathcal{A}|}T}}
> > > > > $$
> > > > > $$
> > > > > \leq  2\sqrt{2}\sqrt{\ln\frac{4|\mathcal{S}||\mathcal{A}|NT}{\epsilon}}{ \sqrt{N^2{|\mathcal{S}|}T}}~~~\text{due to} |\mathcal{A}|=2
> > > > > $$
> > > > > We missed a pre-factor $2$ in the proof, and we modified this typo in the appendix of the paper (changes are highlighted in blue). Thank you again for your careful proofreading.
> > > > >
> > > > >
> > > > > **Question \#6: In the sentence between lines 1398 and 1399, ...**
> > > > >
> > > > >
> > > > > **Response:** Thank you again for your careful proofreading. This has been resolved in the response to your **Question \# 5**, and we modified the appendix of the paper (changes are highlighted in blue) correspondingly.

---

> > > > > > ### Author Response · Authors · 2024-11-25
> > > > > > **Follow-up**
> > > > > >
> > > > > > Dear Reviewer FhNw,
> > > > > >
> > > > > > Since the public discussion phase is ending soon, we just wanted to check in and ask if our rebuttal clarified and answered your questions. We would be very happy to engage further if there are additional questions.
> > > > > >
> > > > > > Also, we wanted to check if our additional clarifications regarding the merits of the paper would convince the reviewer to raise the score. Thank you!
> > > > > >
> > > > > > Best,
> > > > > >
> > > > > > Authors of Paper 5089

---

> > > > > > ### Comment · Reviewer_FhNw · 2024-11-26
> > > > > > **Feedback to rebuttal**
> > > > > >
> > > > > > Thank you for addressing my reviews. Most of the issues have been clarified, but there are still a few questions I would like to discuss further.
> > > > > >
> > > > > > Response to comment for Weakness #2 :
> > > > > > I am afraid there might be a misunderstanding regarding my question. I fully understand how you reduce the total of $B(B-1)/2$ comparisons to $B-1$. My concern is about the third line in Algorithm 3, where, after selecting $B$ arms at each step, you state that $B-1$ duels are performed randomly. When solving (12) in Algorithm 2, you rely on the $j$-th element in the $((*-1)|S|+\sigma_{s^*})$-th column to infer the preference between $s_n$ and $s_*$. This implies that, in this process, only the preference data between $s_n$ and $j$, $j$ and $s_*$, and $s_n$ and $s_*$ are utilized to construct $\hat{\mathbf{F}}^{*,k+1}_n(\sigma_{s_n}, \sigma_{s_*})$ at each round. Consequently, it is very likely that if $s_n$ is compared with another state $i$ in a duel, but the preference data between $s_n$ and $i$ will not be used in this round of preference inference. My question is: does this mean the preference data between $s_n$ and $i$ is wasted at each $k$?
> > > > > >
> > > > > > Furthermore, instead of performing $B-1$ duels randomly, would it be possible to strategically select the duels to ensure that we gather more useful preference data—specifically between $s_n$, $j$, and $\sigma_{s_*}$—which will actually be used in solving (12)?
> > > > > >
> > > > > > Response to comment for Question #1:  Kindly refer to my restated question above regarding Weakness #2.
> > > > > >
> > > > > > Response to comment for Question #3:  I find your statement a bit unclear regarding $Q_n(s)$ representing the expected reward over the entire episode $k$. In your problem formulation (lines 131 to 142), you describe the restless multi-armed bandit (MAB) problem, where, at each time step, selecting arm $n$ in state $s$ results in a reward $r_n(s)$, which is defined as the reward for a single time step. Then, in Proposition 1, you provide the equation $r_n(s) = r_*(s_*) + Q_n(s)$. Given this equation, and since $r_n(s)$ is defined for a single time step, it seems reasonable to conclude that $Q_n(s)$ should also represent the preference-reference at a single time step, rather than over the entire episode.
> > > > > >
> > > > > > Response to comment for Question #5:  Thank you for revising the proof. I believe the new step introduced between lines 1388 and 1389 is a key addition that addresses the missing connection between the preceding and subsequent steps. However, I still have some concerns regarding the steps from lines 1382 to 1386. For example, in the expression from lines 1385 to 1386, if the algorithm visits arm $a$ in state $s$ once at the beginning and never revisits this state, it would result in $Z^1_n(s,a) = \dots = Z^K_n(s,a) = 1$. This would yield a term proportional to $K/\sqrt{2}$ in the expression from lines 1385 to 1386, which scales as $O(T/H)$ and contradicts your final results. In fact, if you simply remove the steps from lines 1382 to 1386 in the proof, it seems to make the argument more consistent.

---

> > > > > > > ### Author Response · Authors · 2024-11-27
> > > > > > > **Further clarification to the follow-up comments**
> > > > > > >
> > > > > > > We are happy to hear that we have addressed most of the issues. Thank you for engaging in the discussion and giving us the opportunity to offer further clarification.
> > > > > > >
> > > > > > > **Q1: ... wasted comparison...**
> > > > > > >
> > > > > > > **Response:** We thank the reviewer for further clarifying the previous comment. Our answer is that the preference value is not wasted. As noted, the randomly selected $B-1$ pairs of duels are a subset of the $(B-1)B/2$ possible pairwise duels. If the reviewer’s claim was valid for the randomly selected $B-1$ duels, it would also hold for the complete $(B-1)B/2$ duels. However, this is not the case. Each duel contributes to improving the estimation accuracy and confidence of a particular element in the preference matrix $\mathbf{F}$. While a specific duel may not be "directly" utilized in the current episode, it enhances the inference process in subsequent episodes. To illustrate, consider the following example:
> > > > > > >
> > > > > > > We have five arms 1, 2, 3, 4, 5, and arm 4 is the reference arm $\star$. At the current step, arms 1, 2, 3, 5 are pulled, resulting in six different pairwise comparisons: (1, 2); (1, 3); (1, 5); (2,3); (2,5) and (3,5). Let us take a random selection as (1, 2); (2, 3), and (3, 5).  The DOPL algorithm updates the estimations for (1, 4), (2, 4), (3, 4), and (5, 4). Suppose the current estimate for (2, 4) has the highest confidence, allowing us to infer the value of (1, 4) using the value of (1, 2). If the inferred value is less reliable than the original (1,4), the algorithm retains the original value. However, in the next episode, another duel (1, 2) may occur. With a significantly higher confidence in (1, 2) by this time, (1, 2) and (2, 4) can jointly provide a highly confident inference of (1,4).
> > > > > > >
> > > > > > > This example highlights the importance of seemingly indirect contributions, as they enhance the inference quality over time, even if their impact is not immediately apparent.
> > > > > > >
> > > > > > > **Q2: ... $h$ in the episode...**
> > > > > > >
> > > > > > > **Response:**  We thank the reviewer for further clarifying this question. Indeed, the current expression of $Term_1$ in Eq. (46) is an average value for each step, which needs to times with $H$. However, this will not affect the final results, due to the relaxation in lines 1378-1380  with the fact that $H\cdot\omega_n^k(s,a)=\sum_{h=1}^{H}\mathbf{1}(s_n(k,h)=s,a_n(k,h)=a)$.
> > > > > > >
> > > > > > > **Q3: ... Lines 1382 to 1386...**
> > > > > > >
> > > > > > > **Response:**  We sincerely thank the reviewer for carefully examining the proof in our paper. Your feedback has significantly contributed to improving the quality of this work. We now fully understand the point raised by the reviewer and have removed the relevant lines accordingly. The updates have been incorporated into our revised proof. An updated draft has been uploaded.

---

### Official Review · Reviewer_xkaa · 2024-11-04

**Soundness:** 3
**Presentation:** 4
**Contribution:** 4
**Rating:** 6
**Confidence:** 3

**Summary:**

The paper studies the Restless Multi-Armed Bandits with preference feedback named PREF-RMAB. The authors propose the Direct Online Preference Learning (DOPL) algorithm achieving an $\tilde{O}(\sqrt{T \ln T})$ regret, which is the first to give the theoretical regret upper bound for PREF-RMAB. Moreover, the paper presents numerical experiments which further validate the efficacy of DOPL against various baselines.

**Strengths:**

1.	The paper successfully integrates preference feedback within the RMAB framework, a novel approach that shifts away from traditional scalar reward dependency. Moreover, the presented algorithm DOPL achieves $\tilde{O}(\sqrt{T \ln T})$ regret with theoretical analysis.
2.	The relaxed LP-based direct index policy of DOPL is also given to tackle the limitations of computational intractability.
3.	The writing is clean and easy to follow.

**Weaknesses:**

1.	Estimating the whole preference matrix $F$ in DOPL algorithm requires large computational cost. Moreover, it would be beneficial to involve a thorough discussion on computational complexity of DOPL.
2.	In experiments, the existing algorithms like MLE_WIBQL, MLE_LP fail to achieve sublinear regret. A detailed discussion on why these algorithms underperform in achieving sublinear regret would provide valuable insights.

**Questions:**

1.	Estimating the whole preference matrix $F$ in DOPL algorithm requires large computational cost. Moreover, it would be beneficial to involve a thorough discussion on computational complexity of DOPL.
2.	In experiments, the existing algorithms like MLE_WIBQL, MLE_LP fail to achieve sublinear regret. A detailed discussion on why these algorithms underperform in achieving sublinear regret would provide valuable insights.

---

> ### Author Response · Authors · 2024-11-20
> **Official Response by Authors (1/2)**
>
> Thank you very much for your review and constructive comments, as well as giving the positive rating of our work. Here we would like to address the reviewer's concerns and hope that can help raise the rating of our paper.
>
> **Weakness \#1:  Estimating the whole preference matrix $F$
>  in DOPL algorithm requires large computational cost. Moreover, it would be beneficial to involve a thorough discussion on computational complexity of DOPL.**
>
> **Response:**
> Thank you for your valuable feedback on the computational complexity of the DOPL algorithm, specifically concerning the estimation of the preference matrix
> $\mathbf{F}$.  We would like to emphasize the following aspects and design choices in our paper that address this "complexity" challenge:
>
> **Only Need to Learn One Column in $\mathbf{F}$.**  It is true that a direct estimation of the entire preference matrix $\mathbf{F}$ would indeed require a substantial computational effort. However, a key observation in this paper (Lemmas 1-2) is that the reward for arm $n$ in any state $s$ can be expressed by the reward of a reference arm $\star$ in a reference state $s_\star$.  Specifically, as we discussed in Remark 2 and Remark 3, the preference of arm $n$ in state $s$ can be well represented by
> $Q_n(s):=\ln\frac{\mathbf{F}_n^\star(\sigma_s, \sigma\_{s\_\star})}{1-\mathbf{F}_n^\star(\sigma_s, \sigma\_{s\_\star})},$
>
> which we call the "preference-reference term". Therefore, to estimate the preference of arm $n$ in state $s$ or the value of $Q_n(s)$, DOPL only needs to learn one specific column of the preference matrix $\mathbf{F}$, i.e., $\mathbf{F}(: ,(\star-1)|\mathcal{S}|+\sigma_{s_\star})$ (see discussions in lines 272-279). This corresponds to the preference between the reference state $s_\star$ of reference arm $\star$ with any arm $n\in\mathcal{N}$ in any state $s\in\mathcal{S}$. As we discussed later (Remark 2 and Lemma 5), the design of our DOPL is regardless of the choice of this reference arm and its reference state.
>
> **Reduction in Computational Complexity through Preference Inference.** In addition, our DOPL introduces a novel preference inference mechanism (Step 3 of Section 3.2) to mitigate the comparison cost. Rather than estimating the preference between all pairs of activated arms in each time slot, DOPL leverages empirical preference estimates from comparisons of a "limited" number of arms and infers the preferences for less frequently observed states using relationships derived from other estimated pairs, as depicted in Lemma 4. Specifically, for any pair $(j_1, j_2)$, the inferred preference $\hat{\mathbf{F}}^{inf}(j_1, j_2)$ is computed using previously established empirical estimates
> $\hat{\mathbf{F}}(j, j_1)$ and $\hat{\mathbf{F}}(j, j_2)$ from comparisons involving an intermediary reference arm
> $j$. The inference step significantly reduces the number of comparisons required to maintain an accurate preference estimation. More specifically, we analyze the complexity of DOPL in terms of the number of comparisons and updates required. A naive estimation approach for a preference matrix involving $B$ activated arms at each time slot would necessitate $O(B^2)$ comparisons. In contrast, the preference inference in our DOPL reduces this requirement to $O(B-1)$ comparisons by leveraging transitive relationships, thereby improving computational efficiency while maintaining robust estimation guarantees.
>
> **Computational Complexity of DOPL.** As discussed in Section 3 (lines 209-213), there are three key components in DOPL: (i) construct confidence set; (ii) online preference learning; and (iii) direct index policy design. The computations mainly come from the second and third components, both of which are linearly in the number of arms and the size of state space. Specifically, the online preference learning (Section 3.2) needs to infer the specific column of preference matrix $\mathbf{F}$ and its computational complexity is linear in the number of arms $N$
> , and the dimension of state space $|\mathcal{S}|$
> . Similarly, the direct index policy design (Section 3.3) needs to solve an LP, for which the computational complexity grows linearly with the number of arms $N$
> , and the dimension of state space
> $|\mathcal{S}|$. Hence, **the computational complexity of DOPL scales well with larger inputs.**

---

> ### Author Response · Authors · 2024-11-20
> **Official Response by Authors (2/2)**
>
> **Weakness \#2:  In experiments, the existing algorithms like MLE_WIBQL, MLE_LP fail to achieve sublinear regret. A detailed discussion on why these algorithms underperform in achieving sublinear regret would provide valuable insights.**
>
> **Response:** Thank you for your observations. We appreciate the opportunity to elaborate on why these algorithms fail to achieve sublinear regret in our experimental settings, providing valuable insights into their limitations in the presence of preference feedback in practice.
>
>
> Note that WIBQL (Whittle-index based Q-learning) and LP (linear programming) were designed for the conventional RMAB setting with scalar rewards. Both approaches are guaranteed to converge asymptotically. However, they are not guaranteed with a finite-time performance bound (i.e., a sublinear regret). To make them compatible with the presence of preference feedback, we introduced MLE_WIBQL and MLE_LP on top of them, i.e., MLE_WIBQL and MLE_LP rely on maximum likelihood estimation (MLE) to convert preference feedback into scalar reward estimates before applying the corresponding WIBQL and LP algorithms. This indirect approach inherently introduces noise and estimation errors during the transformation from preference-based data to scalar rewards. The cumulative effect of these inaccuracies during each decision epoch leads to suboptimal action selection, resulting in higher regret. In addition, converting preference feedback into scalar rewards typically assumes that an accurate mapping function exists. However, in complex dynamic environments, e.g., the RMAB with preference feedback as modeled by our proposed \textsc{Pref-RMAB} framework, such mappings can be non-linear, unknown, or resource-intensive to approximate correctly. Different from these existing methods, our DOPL circumvents this issue by directly learning from preference feedback, eliminating the need for a reward transformation step. This direct approach allows DOPL to leverage preference data more effectively for decision-making, reducing the impact of noise and achieving sublinear regret.

---

> > ### Author Response · Authors · 2024-11-25
> > **Follow-up**
> >
> > Dear Reviewer xkaa,
> >
> > Since the public discussion phase is ending soon, we just wanted to check in and ask if our rebuttal clarified and answered your questions. We would be very happy to engage further if there are additional questions.
> >
> > Also, we wanted to check if our additional clarifications regarding the merits of the paper would convince the reviewer to raise the score. Thank you!
> >
> > Best,
> >
> > Authors of Paper 5089

---

> > > ### Author Response · Authors · 2024-11-30
> > > **Additional Feedback?**
> > >
> > > Dear Reviewer xkaa,
> > >
> > > Once again, thanks for your comments. As the discussion period winds down soon, please follow up if our rebuttal clarified and answered your questions, and if we can answer or clarify additional points.
> > >
> > > Best,
> > >
> > > Authors of Paper 5089

---

### Official Review · Reviewer_j6r7 · 2024-11-06

**Soundness:** 2
**Presentation:** 2
**Contribution:** 2
**Rating:** 5
**Confidence:** 3

**Summary:**

This work studies a new problem set-up, PREF-RMAB.
For me, the problem set-up is very incremental. It is quite similar to duelling bandits. The proposed set-up is more like duelling bandits with state transitions.

The writing needs to be improved.

The writing for the intro is too wordy. I wish to see more literature work discussions.

I suggest putting the main theorem (Theorem 1) earlier. I can only see the theoretical result at Page 8. So, the structures for sections 4 and 5 are suggested to re-arrange.

A minor thing: usually, RL or bandit theory works use $\delta$ to be the failure probability. The greek letter $\epsilon$ is for something else, like the error rate.

I went through the proofs and some steps are novel, not that typical in bandit/RL literature. But I did not check the correctness of the proofs.

**Strengths:**

see the first box

**Weaknesses:**

see the first box

**Questions:**

see the first box

---

> ### Author Response · Authors · 2024-11-20
> **Official Response by Authors (1/3)**
>
> Thank you very much for your review and constructive comments. Here we would like to address the reviewer's concerns and hope that can help raise the rating of our paper.
>
> **Comment 1: ...compare with dueling bandits...**
>
> **Our Response:**  Thank you for your comment, however, we are afraid that we cannot agree with the reviewer on this comment. First of all, we would like to highlight that this paper studies the **restless multi-armed bandits (RMAB)** problem, which dramatically differs from the widely studied multi-armed bandit (MAB) problem, including the dueling bandits, for online decision making. Each ``restless'' arm in RMAB is modeled as an Markov decision process (MDP) with a state evolving according to a transition function, while arms are stateless in conventional MAB or in dueling bandits.  Although RMAB has been widely used to model sequential decision making with **instantaneous/hard constraint**,  it is computationally intractable and PSPACE-hard (Papadimitriou & Tsitsiklis, 1994).  In addition, even compared to the standard RL setting where the agent interacts with a single MDP to optimize cumulative reward, RMAB encompasses multiple MDPs,  coupled through an instantaneous/hard constraint that limits the number of arms the DM can activate at any time slot.
>
> This coupling, due to the shared constraint, introduces a layer of complexity and interdependence absent in single-MDP RL scenarios.  A well-known key technical challenge for RMAB problems is how to develop "low-complexity'' solutions in either offline or online settings for decision makings with the goal to maximize the objective in Eq. (1) while strictly satisfy the instantaneous/hard constraint in each time slot.  See detailed discussions on the challenges below. Hence, our \textsc{Pref-RMAB} is **NOT** ``simply'' *an incremental* setting of the dueling bandit setting with state transitions.
>
> Secondly, it is known that in a dueling bandit setting, the agent/decision maker selects a pair of arms $(k_{+1,t}, k_{-1,t})$ from the set $[N]$ at each time slot $t$ and a preference feedback $\alpha_t \sim \text{Ber}(P_t(k_{+1,t}, k_{-1,t}))$ is revealed, with $P_t(k_{+1,t}, k_{-1,t})$ being the probability of arm $k_{+1,t}$ being preferred over arm $k_{-1,t}$.
> The goal is to find the best arm $k^*$ to minimize the total loss
> measured w.r.t. the single fixed arm $k^* \in [N]$ in hindsight, i.e.,
> \begin{align*}
> R_T(k^*) := \sum_{t=1}^{T} \frac{1}{2} \left( P_t(k^*, k_{+1,t}) + P_t(k^*, k_{-1,t}) - 1 \right).
> \end{align*}
> **In such a dueling bandit setting, it is known that a single best arm can be found by preference feedback only with multiple rounds.**
> In contrast, in our proposed \textsc{Pref-RMAB} setting, the goal is to solve the following problem (derived from Lemma 5):
> $$
> \max_{\mu^\pi} J(\pi):=  \max_{\mu^\pi} \left[\sum_{n\in\mathcal{N}}\sum_{s\in\mathcal{S}}\sum_{a\in\mathcal{A}} \mu_n(s,a)Q_n(s)+\sum_{n\in\mathcal{N}}\sum_{s\in\mathcal{S}}\sum_{a\in\mathcal{A}} \mu_n(s,a)r_\star(s_\star)\right]
> $$
> $$
> \mbox{s.t.} {\sum_{n\in\mathcal{N}}\sum_{s\in\mathcal{S}}\sum_{a\in\mathcal{A}} a\mu_n(s,a)\leq B},
> $$
> $$
> {\sum_{a}} \mu_n(s,a)=\sum_{s^\prime}\sum_{a^\prime}\mu_n(s^\prime, a^\prime)P_n(s^\prime|s,a^\prime),\forall n,
> $$
> $$
> \sum_{s\in\mathcal{S}}\sum_{a\in\mathcal{A}}\mu_n(s,a)=1,
> ~\forall n, s, a,
> $$
>
> with $\star$ being a reference arm, $s_\star$ being a reference state for the reference arm, and $Q_n(s)$ being the ``preference-reference" term defined in Lemma 2.**The goal in \textsc{Pref-RMAB} is not to find a best arm at every time step, but instead to leverage preference feedback to design a sequential decision-making policy for the coupled MDP problem.** To achieve this goal, there are many unique challenges in our proposed \textsc{Pref-RMAB} setting as follows:

---

> > ### Author Response · Authors · 2024-11-20
> > **Official Response by Authors (2/3)**
> >
> > - First, we highlight the unique challenges that must be overcome to design low-complexity solutions for \textsc{Pref-RMAB}. As discussed in Section 3 (lines 204-212), we leverage the UCRL-based algorithm for the online \textsc{Pref-RMAB} and need to address three key challenges: The first challenge is how to handle unknown transitions of each arm. Specifically, we construct confidence set to guarantee the true one lines in these sets with high probability.
> > The second challenge is how to handle unknown preference model. Note that we did not learn a scalar reward function to represent the preference feedback of human as in RLHF and then apply existing RL methods for RMAB.  Instead, we **directly** leverage the preference feedback to make decisions in online \textsc{Pre-RMAB}.  A key contribution in this paper to handle this challenge is that we develop the  ''preference reference'', which not only can infer the empirical preference of arms that are not visited frequently, but also with a bounded inference error  (see details in ''Step 3'' in lines 301-328).
> > The third challenge is how to handle the ``instantaneous'' constraint in \textsc{Pref-RMAB}. As in existing RMAB literature, one way to address this is via developing low-complexity index policy.  A key contribution in this paper is that we show that we can directly define a linear programming for \textsc{Pre-RMAB} in terms of preference feedback (see Section 3.3).
> >
> > - Second, these new design challenges for \textsc{Pre-RMAB} are further reflected in the regret analysis of our proposed DOPL algorithm. Specifically, we propose a new regret decomposition (Lemma 6) to tie the regrets to the online preference learning and the direct index policy.  Their regrets and the analysis challenges are discussed in Lemma 7 and lines 436-442, and Lemma 8 and lines 444-448, respectively. Finally, we would like to highlight that our DOPL is the first to achieve a sub-linear regret for \textsc{Pre-RMAB}, which matches that of the standard RMAB with standard rewards, that is already very challenging (again RMAB is different from conventional MAB problems).
> >
> > Given all the distinct challenges and setting differences, our proposed PREF-RMAB is **NOT** ``simply'' *an incremental* setting of the dueling bandit setting with state transitions.

---

> > > ### Author Response · Authors · 2024-11-20
> > > **Official Response by Authors (3/3)**
> > >
> > > **Comment \#2:  The writing for the intro is too wordy. I wish to see more literature work discussions.**
> > >
> > > **Response:** We appreciate your suggestion to provide more focused discussions of related literature. First of all, we  provided many related works on RMAB (lines 37-44), an area that is most related to this work. However, to our best knowledge, all  existing RMAB works considered the settings where the decision makers rely on the absolute scalar rewards feedback, while this paper introduces a novel setting that is the first of its kind. In other words, we introduced \textsc{Pref-RMAB}, where the decision maker only receives preference-feedback, which contains arguably less information than scalar rewards, making our \textsc{Pref-RMAB} more difficult.
> > > Thus, our primary aim in the introduction is to emphasize and motivate the significance of this particular setting. Due to space constraint, for a comprehensive discussion of related literature, including works on dueling bandits, reinforcement learning from human feedback (RLHF), and conventional RMAB, we have provided a detailed review in the related work section.
> > >
> > > **Comment \#3:  I suggest putting the main theorem (Theorem 1) earlier. I can only see the theoretical result at Page 8. So, the structures for sections 4 and 5 are suggested to re-arrange.**
> > >
> > > **Response:** Thank you for your suggestion, and we appreciate it, especially your recognition of our theoretical contributions in this paper. As we responded above to your first comment, there are many challenges and open problems in designing low-complexity solutions for \textsc{Pref-RMAB}. To this end, the first set of contributions in this paper is the design of **DOPL** that addresses these challenges, which are described in Section 3. Once **DOPL** is designed, the second set of contributions in this paper is that we prove a sub-linear regret for **DOPL**, which matches that of the standard RMAB with scalar rewards (see discussions in Remark 7). Per the reviewer's suggestion, we further improve the Introduction (changes are highlighted in blue), i.e., we make some statements much more concise, and (informally) introduce the result of Theorem 1 in the last paragraph of the Introduction with some detailed discussions on the importance of this result.

---

> > > > ### Author Response · Authors · 2024-11-25
> > > > **Follow-up**
> > > >
> > > > Dear Reviewer j6r7,
> > > >
> > > > Since the public discussion phase is ending soon, we just wanted to check in and ask if our rebuttal clarified and answered your questions. We would be very happy to engage further if there are additional questions.
> > > >
> > > > Also, we wanted to check if our additional clarifications regarding the merits of the paper would convince the reviewer to raise the score. Thank you!
> > > >
> > > > Best,
> > > >
> > > > Authors of Paper 5089

---

> > > > > ### Author Response · Authors · 2024-11-30
> > > > > **Additional Feedback?**
> > > > >
> > > > > Dear Reviewer j6r7,
> > > > >
> > > > > Once again, thanks for your comments. As the discussion period winds down soon, please follow up if our rebuttal clarified and answered your questions, and if we can answer or clarify additional points.
> > > > >
> > > > > Best,
> > > > >
> > > > > Authors of Paper 5089

---

### Meta-Review · Area_Chair_XDf8 · 2024-12-20

**Metareview:**

The paper introduces PREF-RMAB, a new variant of the Restless Multi-Armed Bandits (RMAB) problem that leverages preference feedback instead of direct reward estimations. This approach is particularly relevant for applications such as app marketing and CPAP treatment, where obtaining exact rewards can be challenging. PREF-RMAB differs from traditional dueling bandits by incorporating state transitions, making it a more complex and realistic model.

To address PREF-RMAB, the authors propose the Direct Online Preference Learning (DOPL) algorithm, which is the first to establish a theoretical regret upper bound for this setup. The algorithm operates by expressing the reward function of each arm in any state as a combination of a reference reward and a preference-based function. This allows the development of a direct index policy that selects which arms to activate based on preference data in an online manner. Additionally, the authors introduce a method to infer the empirical average of arm preferences by utilizing the preferences of other arms, effectively addressing the issue of infrequent visits to certain arms or states.

Theoretical analysis confirms the efficacy of DOPL, and numerical experiments demonstrate its improved performance compared to baseline methods. The paper also discusses the transformation of the original reward-based optimization problem into a preference-based one, enabling a low-complexity decision-making process.

Overall, the study makes a contribution to the field of RMAB. The reviewers have been thorough in assessing different aspects of the paper. They all comment on the solid contribution of the paper to the restless bandit literature. On the other hand, they all share the concern that the presentation could have been clearer (less verbose in places and better organized). Overall, the intellectual merits of the paper outweigh its weaknesses, and I recommend this paper for publication.

**Additional Comments On Reviewer Discussion:**

There were constructive discussions among the authors and reviewers, and overall, the authors' responses have clarified some of the reviewers' concerns, leading to a near consensus among the reviewers about their rating of the paper.

---

### Decision · Program_Chairs · 2025-01-22

Accept (Poster)